



# Identification of snowfall microphysical processes from vertical gradients of polarimetric radar variables

Noémie Planat[1,2], Josué Gehring[1], Étienne Vignon[1,3], and Alexis Berne[1]

[1]Environmental Remote Sensing Laboratory (LTE), École Polytechnique Fédérale de Lausanne, Lausanne, Switzerland
[2]Department of Atmospheric and Oceanic Sciences, McGill University, Montréal, Québec, Canada
[3]Laboratoire de Météorologie Dynamique, Institut Pierre-Simon Laplace, Sorbonne Université / CNRS / École Normale Supérieure – PSL Research University / École Polytechnique – IPP, Paris, France

**Correspondence:** Alexis Berne (alexis.berne@epfl.ch)

**Abstract.** Polarimetric radar systems are commonly used to study the microphysics of precipitation. While they offer continuous measurements with a large spatial coverage, retrieving information about the microphysical processes that govern the evolution of snowfall from the polarimetric signal is challenging. The present study develops a new method, called Process Identification based on Vertical gradient Signs (PIVS), to spatially identify the occurrence of the main microphysical processes
(aggregation and riming, crystal growth by vapor deposition, and sublimation) in snowfall from dual-polarization Doppler radar scans. We first derive an analytical framework to assess in which meteorological conditions the local vertical gradients of radar variables reliably inform about microphysical processes. In such conditions, we then identify regions dominated by *(i)* vapor deposition, *(ii)* aggregation and riming and *(iii)* snowflake sublimation and possibly snowflake breakup based on the sign of the local vertical gradients of the reflectivity $Z_H$ and the differential reflectivity $Z_{DR}$. The method is then applied to
data from two frontal snowfall events: one in coastal Adélie Land, Antarctica and one in the Taebaeck mountains in South Korea. The validity of the method is assessed by comparing its outcome with snowflake observations using a Multi-Angle Snowflake Camera and with the output of a hydrometeor classification based on polarimetric radar signal. The application of the method further makes it possible to better characterize and understand how snowfall forms, grows and decays in two different geographical and meteorological contexts. For the Antarctic case study, we show that crystal growth by vapor deposition
dominates above 2500 m a.g.l., aggregation and riming prevail between 1500 and 2500 m a.g.l. and snowflake sublimation by low-level katabatic winds occurs below 1500 m a.g.l.. For the event in South Korea, aggregation and riming dominate between 4000 and 4800 m a.g.l., with local sublimation below and vapor deposition above. We infer some microphysical characteristics in terms of radar variables from statistical analysis of the method output (e.g. $Z_H$ and $Z_{DR}$ distribution for each process). We finally highlight the potential for extensive application to cold precipitation events in different meteorological contexts.

# 1 Introduction

The characterization and modeling of snowfall necessitate a thorough understanding of the dynamical and microphysical processes driving snowflakes growth and evolution from the synoptic to the micro-scale (e.g., Ryzhkov and Zrnić, 2019; Morrison et al., 2020). Identifying and quantifying microphysical processes require the collection of accurate observations. Despite solid





precipitation being generally frequent in polar climates and mountainous regions, ground-based or in-situ measurements of

clouds and precipitation remain limited in such areas due to the harsh meteorological conditions and sparsity of meteorological stations. Moreover, few methods and metrics exist to systematically and operationally give access to the occurrence and properties of microphysical processes in mountainous and polar snowfall.

Ground-based remote sensing instruments make it possible to gain insights into precipitation formation and vertical evolution

in remote areas at high altitude and latitude (e.g., Shupe et al., 2011; Grazioli et al., 2015; Gorodetskaya et al., 2015; Wen et al., 2016; Grazioli et al., 2017a; Scarchilli et al., 2020; Lubin et al., 2020).

The added value of radar polarimetry (e.g., Kumjian, 2012; Tiira and Moisseev, 2020) is of particular relevance for characterizing salient features in radar profiles like the melting layer (Griffin et al., 2020) but also for deciphering the microphysical structure of snowfall. Bader et al. (1987), Andrić et al. (2013), Schneebeli et al. (2013), Moisseev et al. (2015), Grazioli et al.

(2015), Besic et al. (2018), Gehring et al. (2020b) among others show the potential of dual-polarization variables to study some characteristics of snowflakes (e.g., type, shape, orientation, size, phase) as well as the underlying microphysical processes that drive their evolution.

However the complex microphysical properties of snowflakes (i.e. large variety in density, size and shape) hamper an accurate and universal retrieval of their physical and geometrical properties from polarimetric radar data. Various studies used

different polarimetric and Doppler signatures to identify the main cold processes (aggregation, vapor deposition, sublimation and riming) depending on the context. For instance, the onset of aggregation was identified by Moisseev et al. (2015) as bands of high values of specific differential phase $K_{dp}$. Kennedy and Rutledge (2011); Gehring et al. (2020b) identified a region dominated by aggregation with a decrease in differential reflectivity ($Z_{DR}$) collocated with an increase of reflectivity ($Z_H$) while Schrom and Kumjian (2016) identified aggregation as a maximum in the downward relative gradient of reflectivity $\partial_z Z_H$

below the $-15°$C isotherm and a positive downward gradient of absolute Doppler velocity $\partial_z |V_r| > 0$.

Various methods have also been developed to extract information from the spatial and temporal evolution of radar polarimetric variables. The Quasi Vertical Profiles (QVPs) method was introduced by Ryzhkov et al. (2016). QVPs are computed from Plan Position Indicator (PPI) radar scans at different elevation angles that are azimuthally averaged to give vertical profiles of polarimetric variables above the radar. A similar method named Columnar Vertical Profiles (CVPs) provides vertical profiles

at any point of space and was presented in Murphy et al. (2020). The authors emphasize that the noise and standard deviation of the signal are strongly reduced owing to the azimuthal averaging. Range Height Indicator (RHI) scans can also be used to extract vertical profiles at a given azimuth (e.g., Andrić et al., 2013; Moisseev et al., 2015; Gehring et al., 2020b). The analysis of vertical profiles of dual-polarization variables from RHI scans has been further developed by Tiira and Moisseev (2020) with a multivariate unsupervised classification of vertical profiles of $Z_H$, $Z_{DR}$ and $K_{dp}$. The mean profiles of the resulting

classes have been a posteriori interpreted in terms of specific meteorological conditions and microphysical processes.

One should however be careful when interpreting the vertical profiles obtained with those methods in conditions of strong wind shear. Indeed, strong wind shear leads to slanted streamlines of snowflakes thereby manifesting as clear fallstreaks in RHI scans or time-height plots. In such cases, the shape of the vertical profiles of radar variables strongly depends upon advection





mechanisms and not only on microphysics. Kalesse et al. (2016) discuss this issue by characterizing the hydrometeor proper-
ties through the analysis of gradients of radar variables along the pre-identified fallstreaks during a snowfall event in Finland.
However, in which conditions one can reliably associate the local vertical profiles of polarimetric variables to the occurrence
or intensity of a given microphysical processes in snowfall is still, to our knowledge, an open question.

In the present study, we propose a new method to automatically identify the microphysical processes in snowfall from
dual-polarization radar quantities. This method, referred hereafter to as PIVS (Process Identification based on Vertical gradi-
ent Signs), is systematic and based on vertical gradients of $Z_H$ and $Z_{DR}$, therefore focusing on the vertical evolution of the
particles characteristics. PIVS will help characterize the occurrence and properties of the dominant microphysical processes
governing the snowfall evolution in space and time. A preliminary step consists in developing an analytical framework to theo-
retically determine the conditions in which the vertical analysis of polarimetric radar signal provides robust information about
snowfall microphysics. This method is then illustrated over two case studies to identify and characterize the microphysical pro-
cesses at play: one case at Dumont d'Urville (DDU) station, Adélie Land, Antarctica and one case in the Taebaeck mountains,
South Korea. As the two events correspond to different hemispheres and different meteorological and orographic conditions,
their analysis will help illustrate the range of application of the designed method.

The paper is structured as follows. We first derive and discuss the analytical framework in Sect. 2. We then present the
development of our method and its implementation in Sect. 3. Section 4 illustrates the method application over the two case
studies and discuss the outcome and the associated limitations. We then summarize our key findings and draw the conclusions
in Sect. 5.

## 2 When do microphysics drive vertical variability?

The vertical structure of radar polarimetric variables strongly depends on how microphysical processes modify the properties
of the crystals and snowflakes. As mentioned hereabove, different studies have exploited the vertical profiles of polarimetric
radar variables to identify and characterize microphysical processes. One may nonetheless question such methodologies in
conditions during which additional mechanisms alter the evolution of radar variables along the vertical direction, the most fre-
quent being advection effects in strong wind conditions or in the presence of fallstreaks. Analyses of microphysical processes
along fallstreaks are very relevant and give insights into the microphysics of complex precipitation systems (e.g., Pfitzenmaier
et al., 2018). However, fallstreak retrieval algorithms are based on the accurate acquisition of the 3D wind field which is often
not available from measurements. Since the proposed method of snowfall microphysical process characterization will be based
on the interpretation of local vertical gradient in Eulerian vertical profiles of polarimetric radar variables (see Sect. 3), it is
hence crucial to clearly define the conditions in which such gradients give access to reliable information about microphysical
processes.



Let us first consider the continuity equation for radar reflectivity $Z_H$ (Passarelli, 1978; Milbrandt and Yau, 2005) (note that similar developments also hold for other radar variables, such as $Z_{DR}$) :

$$\partial_t Z_H + \boldsymbol{\nabla} \cdot (Z_H \boldsymbol{u}) = \sum_{process} \mathrm{d}_t Z_H|_{process} + \partial_z (v_z Z_H) \tag{1}$$

where $z$ denotes the vertical coordinate oriented upward, $v_z$ is the reflectivity-weighted sedimentation velocity vector, and $\boldsymbol{u}$ is the wind vector whose respective components in the $(x, y, z)$ directions are $(u, v, w)$. $\sum_{process} \mathrm{d}_t Z_H|_{process}$ represents all the microphysical source/loss terms for $Z_H$ while the rightmost term of the equation is the particle sedimentation term. Once developed, this equation reads:

$$\partial_t Z_H + Z_H \partial_x u + u \partial_x Z_H + Z_H \partial_y v + v \partial_y Z_H + Z_H \partial_z (w - v_z) + (w - v_z) \partial_z Z_H = \sum_{process} \mathrm{d}_t Z_H|_{process} \tag{2}$$

With no loss of generality, one can work in a 2D framework along the horizontal wind direction (noted $x$ for convenience here). Equation (2) hence reads:

$$\partial_t Z_H + Z_H \partial_x u + u \partial_x Z_H + Z_H \partial_z (w - v_z) + (w - v_z) \partial_z Z_H = \sum_{process} \mathrm{d}_t Z_H|_{process} \tag{3}$$

One can notice that the reflectivity source/loss terms $\sum_{process} \mathrm{d}_t Z_H|_{process}$ can be directly related to the vertical gradient of reflectivity - multiplied by the relative vertical velocity of the flow with respect to hydrometeors - $(w - v_z)\partial_z Z_H$ if the other

terms in the equation are negligible.

In what follows, we thus need to assess in which conditions the equality

$$(w - v_z)\partial_z Z_H \approx \sum_{process} \mathrm{d}_t Z_H|_{process} \tag{4}$$

is verified. Let us note $\bar{U}$ (resp. $\bar{Z}$ and $\bar{W}$) the characteristic value of the horizontal wind velocity $u$ (resp. of $Z_H$ and of relative vertical velocity $w - v_z$). $L_{d,X}$ refers to the scale of variation of the variable $X$ in the direction $d$. To verify Eq. (4), it is

sufficient to satisfy the three following conditions:

*Condition 1:* The horizontal divergence term is negligible with respect to the vertical divergence and sedimentation i.e. the vertical evolution of polarimetric variables is not affected by horizontal variations. This condition mathematically reads:

$$|Z_H \partial_x u + u \partial_x Z_H| << |Z_H \partial_z (w - v_z) + (w - v_z)\partial_z Z_H| \tag{5}$$

which, after approximating a derivative by the ratio of typical scales and removing $\bar{Z}$, turns into:

$$\frac{\bar{U}}{L_{x,u}} + \frac{\bar{U}}{L_{x,Z_H}} << \frac{\bar{W}}{L_{z,w-v_z}} + \frac{\bar{W}}{L_{z,Z_H}} \tag{6}$$





*Condition 2:* The vertical divergence of $(w - v_z)Z_H$ is mainly driven by the variation of $Z_H$ itself and not by variations in relative vertical velocity:

$$|Z_H \partial_x (w - v_z)| << |(w - v_z)\partial_z(Z_H)| \tag{7}$$

or in terms of typical scales:

$$\frac{1}{L_{z,w-v_z}} << \frac{1}{L_{z,Z_H}} \tag{8}$$

*Condition 3:* The system is quasi-stationary. In other words, when this condition is verified, the time scales of variations due to microphysical processes are comparable with the typical time scale associated with particle sedimentation in such a way that the vertical profile of reflectivity remains always close to the equilibrium solution:

$$|\partial_t Z_H| << |(w - v_z)\partial_z Z_H| \tag{9}$$

or in terms of variable dimension:

$$\frac{1}{L_{t,Z_H}} << \frac{\bar{W}}{L_{z,Z_H}} \tag{10}$$

When respected, Eq. (6), Eq. (8) and Eq. (10) thus express sufficient physical conditions in which the local vertical gradient of $Z_H$ (times the relative vertical velocity) is thereby directly related to the dominant microphysical processes and can be used to identify the occurrence and the properties thereof. The same approach can be used for the vertical gradient of $Z_{DR}$.

## 3 Development of the PIVS method

We can now introduce the so-called PIVS method to identify the occurrence of microphysical processes in snowfall from the vertical gradients of polarimetric radar variables. The different steps of the method are successively described hereafter and are schematically represented in Fig. 1. It is worth noting that our method is built on $Z_H$ and $Z_{DR}$ vertical gradients only, because these two polarimetric variables show strong signatures in the two events studied in Sect. 4 - as opposed to $K_{dp}$ for which no pattern was observed in EV1 - and because we aim to keep the method simple and robust at first. These two variables will make it possible to perform a process-classification into three main groups and we discuss in Sect. 3.2 a possible method extension using additional polarimetric variables. Moreover, note that $Z_H$ and $Z_{DR}$ gradients are not affected by possible miscalibrations.

The method consists of three main steps. We first present how to extract vertical profiles of radar variables from RHI scans and then we explain how to identify processes from the analysis of vertical gradients of $Z_H$ and $Z_{DR}$. We finally develop the statistical framework to analyse and interpret the results of the method.



**Figure 1.** Schematic of the PIVS method. The two top panels are examples of RHI scans of $Z_H$ and $Z_{DR}$



### 3.1 Step 1: Vertical profiles extraction in RHI scans

The first step aims to extract vertical profiles of $Z_H$ and $Z_{DR}$ from 2-dimensional RHIs. We want the resulting profiles to correctly capture the vertical evolution of the snowfall and contain the microphysical information without being too sensitive to the parameters involved in the extraction procedure.

#### 3.1.1 Selection of *"non-empty"* RHIs

We first select the RHIs with enough signal to be robustly used as input for the PIVS algorithm. We keep the RHIs in which the percentage of occupied gates in an area covering $D_x$ km in horizontal range and $D_z$ km in vertical exceeds a minimum threshold of $p\%$. The $D_x \times D_z$ area is chosen considering the sensitivity of the radar ($D_x$ should delimit the area where the sensitivity is high enough), the vertical extent of the precipitating clouds (limiting $D_z$), the potential presence of a melting layer (potential lower vertical limit, since the method is not applicable in and below the melting layer, see herebelow) and the orography. In conditions where the orography or the nature of the surface is very different between the left and the right part of the RHI, one should consider treating the two different RHI sides independently.

#### 3.1.2 Time averaging of the RHIs

Once RHIs have been properly selected, we apply a time averaging to smooth part of the noise out. Microphysical processes have typical time scales ranging from a few minutes to a few hours. In order to remove noise and keep the microphysical signal, the chosen averaging time (hereafter $\Delta t$) should thus be lower than a few tens of minutes. Because our radar provides RHIs every five minutes approximately (see Sect. 4.1), averaging two consecutive RHIs is a reasonable trade-off.

#### 3.1.3 Horizontal division of the RHIs and vertical profile extraction

One can now extract the vertical profiles from the RHIs. RHIs are initially acquired in an $elevation \times range$ grid that we project onto a $75 \times 75$ m$^2$-resolution Cartesian grid (note that 75 m is the range resolution of our radar, see Sect. 4.1), then we select columns of signal. To further reduce the noise, we also apply a horizontal spatial averaging and extract the median column among neighbouring columns. This step is delicate since the spatial variability of the microphysical processes has to be conserved. The horizontal average distance is noted $\Delta x$ and is typically chosen between $5\%$ to $10\%$ of $D_x$. Profiles are then extracted every $\Delta x/2$.

#### 3.1.4 Selection of usable profile sections and vertical smoothing

We then select the relevant sections of the vertical profiles and perform final processing procedures to obtain a consistent dataset.





- In the top part of the extracted profiles, there might be only a few gates containing significant signal (Signal to Noise Ratio above 0 dB). We calculate and keep the median vertical profile at a certain height only if at least 70% of the gates contain significant signal.

175     – One- and two-gate signal gaps, mostly located at the top of the profiles, are replaced with linearly-interpolated values from the two nearest gates. Then we only keep the parts of the profiles that contain more than six gates with significant signal.

- Finally, we apply a vertical three-gate window moving average to reduce gate-to-gate noise.

## 3.2   Step 2: Process identification

180   Once profiles have been extracted from RHI scans, we compute along each of them the local gradients of reflectivity $Z_H$ and differential reflectivity $Z_{DR}$ at each altitude. Based on Eq. (4), we use the sign of the those gradients to identify the dominant process - i.e. the process that predominantly drives the local evolution of reflectivity or differential reflectivity - at a given altitude.

    Based on past literature, we set criteria to assign a given $Z_H$ and $Z_{DR}$ local vertical gradient sign configuration to a given 185   microphysical process. It is worth noting that the following criteria only hold in snowfall and are not valid for processes within or below the melting layer. Importantly, such criteria are valid only in situations in which hydrometeors have a negative absolute vertical velocity. In strong updrafts i.e. in case of positive net vertical velocity, a given microphysical process will manifest in the radar profile with an opposite vertical gradient sign (see Eq. (4)). It is worth emphasizing that the PIVS method is based only on the sign of the gradients, regardless of their magnitude. *As in Sect. 2, we adopt the convention that the vertical axis is* 190   *oriented upward*.

- Aggregation and riming (hereafter AR) correspond to an increase in reflectivity due to an increase in particle size and/or density with decreasing altitude ($\partial_z Z_H < 0$) and a decrease with decreasing height in $Z_{DR}$ ($\partial_z Z_{DR} > 0$) as particles become more spherical (less oblate) (Li et al., 2018; Ryzhkov and Zrnić, 2019). Riming is a complicated process regarding $Z_{DR}$ mostly due to the counter-effects of decreasing oblateness and increasing density. The dominant effect was reported 195   to be an overall decrease in $Z_{DR}$ (Ryzhkov and Zrnić, 2019). $Z_{DR}$ thus behaves slightly differently for aggregation and riming, but to our knowledge, it is complicated to differentiate them only from the sign of the vertical gradients of $Z_H$ and $Z_{DR}$. We therefore decided to group these two processes together.

- Crystal growth by vapor deposition (hereafter CG) corresponds to an increase in particle size with decreasing height ($\partial_z Z_H < 0$) and an increase in oblateness with decreasing height ($\partial_z Z_{DR} < 0$) as particles generally grow along their 200   longest dimension (Schneebeli et al., 2013; Andrić et al., 2013; Grazioli et al., 2015)

- Sublimation (hereafter SUB) corresponds to a downward relative decrease in particle size and concentration ($\partial_z Z_H > 0$, Grazioli et al., 2017b). Because we define SUB only based on $\partial_z Z_H$, it may potentially include other processes that diminish the radar reflectivity (notably the breakup of snowflakes, see Ryzhkov and Zrnić, 2019).



Aggregation, riming, crystal growth and sublimation do not form an exhaustive list of snowfall microphysical processes. Future
improvements of the method may consider the variations in $Z_{DR}$ also in the SUB category and/or evolution of $K_{dp}$ and cross-correlation coefficient $\rho_{hv}$. This could help distinguish riming/aggregation and identify more processes, notably secondary ice generation processes (very active in Antarctica, see Sotiropoulou et al., 2020) that can be associated with strong signatures in $K_{dp}$ (Sinclair et al., 2016).

### 3.3  Step 3: Data analysis and visualization

To visualize and analyze the 3-dimensional (height, horizontal distance to radar, time) output of the method, and besides inspecting each RHI individually, we employ time-height plots and statistical distributions to synthesize the information. Time-height plots are built as follows. For each altitude range and each time step, we compute the proportion of each microphysical process among the extracted vertical profiles in the horizontal direction within the corresponding RHI scan. We hence estimate the dominant process at a given time and at a given height as the process showing the largest occurrence in the horizontal direction within the RHI. Regarding statistical distributions, we compute the empirical distributions of different variables (e.g., occurrence height, magnitude of the vertical gradient) conditioned or not to a specific process. The large number of profiles in the considered case studies (see Sect. 4) ensures the robustness of the derived statistics.

## 4  Application of the methodology to two case studies

We now present two case studies of snowfall events: one over the coast of the Antarctic continent (hereafter EV1) and one over South Korea (hereafter EV2). The two considered precipitation events are produced by stratiform clouds associated with the passage of a warm front above the location of interest. The meteorological and geographical conditions are however quite different particularly in terms of synoptic moisture advection and effects of the orography and of low-level flows on precipitation. Hence we can test, appreciate and discuss our method during two contrasted snowfall events.

### 4.1  Campaigns and datasets

#### 4.1.1  APRES3 campaign in coastal Adélie Land, Antarctica (EV1)

The first precipitation event was sampled in the framework of the Antarctic Precipitation, Remote Sensing from Surface and Space campaign (APRES3) that took place at the french Antarctic station Dumont d'Urville (DDU), coastal Adélie Land, Antarctica. DDU station is set up on the Petrels Island, $-66.66°$S, $140.00°$E, 41 m a.s.l., $UTC + 10$, 5 km off the landfall of the ice sheet.

One noticeable weather feature at DDU is the low-level katabatic flow blowing down from the high Antarctic Plateau with an easterly to south-easterly direction. This strong and dry flow, vertically extending up to $\approx 1.5$ km, significantly diminishes the amount of precipitation that reaches the ground by sublimating the snowflakes (Grazioli et al., 2017b; Durán-Alarcón et al., 2019; Vignon et al., 2019b).





The APRES3 campaign was heavily instrumented during austral summer 2015-2016 (Grazioli et al., 2017a; Genthon et al.,
2018). Of particular relevance for our work, an X-band dual-polarization Doppler radar (called MXPol, Schneebeli et al., 2013;
Scipión et al., 2013) was deployed. It scanned the atmosphere with a range resolution of 75 m, a maximal range of 30 km and a
Nyquist velocity of $39.8 \ \mathrm{ms^{-1}}$. The scan strategy for EV1 was a repeating succession of two PPI at low elevations, one vertical
profile and two RHI scans approximately parallel and perpendicular to the prevailing katabatic wind direction. This total scan
strategy took about 5 minutes. In the data processing, we have only kept the parts of the RHIs for which the elevation angle
is greater than $5°$ - to avoid ground clutter - and lower than $45°$ - to conserve reliable polarimetric signal - and for which the
altitude with respect to the radar is greater than 500 m to avoid near-field effects. The results presented in this study are from
the $203°$ RHI scans, the more parallel to the direction of the large-scale moisture advection.

The semi-supervised hydrometeor classification algorithm of Besic et al. (2016) has also been applied on the MXPol RHI scans
to determine the hydrometeors' nature from the polarimetric signal. Using the additional demixing module from Besic et al.
(2018) makes it possible to further infer the proportions of different hydrometeor types within a given radar sampling volume.

In addition to MXPol, a Multi-Angle Snowflake Camera (MASC, Garrett et al., 2012) was deployed. Hereinafter, we use the
products of the hydrometeor classification from MASC images developed by Praz et al. (2017). A preliminary processing step
following Schaer et al. (2020) removes blowing snow particles from the dataset.
The precipitation event of interest took place between 28 and 30 December 2015. It was associated with the passage of
a warm front of an extra-tropical cyclone setting at the west of DDU. This type of precipitation system is typical for DDU
(Jullien et al., 2020). The accumulated precipitation at the ground during the event was 3.4 mm (Vignon et al., 2019a).

### 4.1.2 ICE-POP campaign in South Korea (EV2)

The second case study took place on 28 February 2018 in the Taebaeck mountains, near Pyeongchang, in South Korea. It was
an intense precipitation event (55 mm of equivalent liquid precipitation) associated with a warm conveyor belt (WCB, i.e. a
warm and moist airstream ascending along the cold front of an extratropical cyclone, see Green et al. (1966), Harrold (1973),
Browning et al. (1973)). The event was sampled by the suite of instruments deployed during the International Collaborative
Experiments for Pyeongchang 2018 Olympic and Paralympic winter games (ICE-POP 2018) and thoroughly studied in Gehring
et al. (2020b).
Two main sites at different altitudes and locations were instrumented (see Fig.1 in Gehring et al., 2020b). The lowest site (66 m
a.s.l.) was the Gangneung Wonju national University (GWU) close to the shore. The second, Mayhills site (MHS), was located
more inland in the Taebaeck massif at 789 m a.s.l.

The same MXPol radar as during the Antarctic APRES3 campaign was deployed at GWU with a scan strategy adapted
to the orography and the location of the two Korean measurement sites. RHIs scans with a range resolution of 75 m and an
horizontal range of 27.2 km were performed towards MHS at two slightly different azimuths ($225°$ and $235°$). The total scan
strategy lasted 10 min with repetition in the main RHI directions every 5 min. As for the APRES3 campaign, we processed





the RHIs in a way to keep only the part for which elevation angles are $\geq 5°$ and $\leq 45°$. Moreover, due to important ground echos, we will analyze the signal only above an altitude of $2000$ m a.g.l.. At the MHS site, the MASC collected images of
falling snowflakes, while a 94 GHz W-band Doppler cloud profiler (WProf, Küchler et al., 2017) provided high-resolution measurements of the reflectivity and of the full Doppler spectra with a Nyquist velocity of $5.1$ m$s^{-1}$ (dealiased, see Gehring et al., 2020a, for details).

## 4.2 Applicability of the method and implementation

Before applying the method to the case studies, we must verify that the environmental conditions derived in Sect. 2 are fulfilled
so that the local vertical gradients are mostly governed by - and subsequently reflect - the microphysics. More precisely, Eq. (6), (8) and (10) should be satisfied. For this purpose, we need to estimate the characteristic scales $\bar{U}$, $\bar{W}$, $L_{t,Z_H}$, $L_{x,u}$, $L_{x,Z_H}$, $L_{z,Z_H}$, $L_{x,Z_{DR}}$, $L_{z,Z_{DR}}$ and $L_{z,w}$ during the two events.

$\bar{U}$ is obtained by calculating the mean wind velocity from radiosoundings between $500$ m (EV1) or $2000$ m (EV2) and $D_z$ a.g.l. during the events (see Vignon et al., 2019a, b; Gehring et al., 2020b for details on radiosonde data acquisition and
processing). $\bar{W}$ is measured from vertical Doppler velocity measurements. We use MXPol data for EV1 while we prefer using WProf measurements for EV2 given its higher temporal frequency. Note that the vertical Doppler velocity gives directly access to the net (particles size distribution weighted) sedimentation velocity $(w - v_z)$.

Following Skøien et al. (2003), we then determine the characteristic space scales and timescales by means of variogram estimated from radar RHI scans. The variograms are normalized by the variance of the field along the direction we use to
compute the variogram (see Lebel and Bastin, 1985). Skøien et al. (2003) quantify the characteristic length (or time) scales in non-stationary fields by visually localizing the first significant slope change - indicating the shortest decorrelation scale - in variograms plotted with logarithmic axes. As the wind and radar 4D fields are generally non-stationary (in a statistical sense), we follow the same approach except that in our case, the spatial extent of $Z_H$ and $Z_{DR}$ are not sufficient to properly work with logarithmic axes.

Note however that the horizontal variations of the wind, $Z_H$ and $Z_{DR}$ can occur over typical distances that exceed the visibility range of the radar (10 to 15 km for EV1, 15 to 20 km for EV2). This prevents a robust estimation of the horizontal decorrelation distances. Therefore, we estimate the horizontal characteristic length scales from numerical simulations carried out with the Weather Research and Forecasting model (WRF, see Appendix A). As the X-band radar reflectivity and differential reflectivity are not standard variables of WRF, we calculate the variogram of the sixth moment of the snow particle size
distribution $M_S^6$ (which the radar is the most sensitive to). In fact under simplifying assumptions (namely constant density and spherical shape), the radar reflectivity in the Rayleigh regime is proportional to the sixth moment of the particle size distribution of snow particles.

Because the original dataset is 3-D (4-D for WRF dataset), we extract variograms (along any direction) that are also 3-D (4-D). Therefore, the mean and percentiles displayed (as well as the calculation of the decorrelation length) are computed
based on the averaged signal along one (or two) dimension(s) (unless specified, we prefer to average temporally the signal).





| variable | EV1 | EV2 |
|---|---|---|
| $\bar{U}$ [ms$^{-1}$] | 12 | 22 |
| $\bar{W}$ [ms$^{-1}$] | 0.8 | 0.6 |
| $L_{x,u}$ [km] | 45 | >50 |
| $L_{x,Z_H}$ [km] | >15 | >15 |
| $L_{z,Z_H}$ [km] | 0.6 | 0.4 |
| $L_{x,Z_{DR}}$ [km] | >15 | >15 |
| $L_{z,Z_{DR}}$ [km] | 0.5 | 0.3 |
| $L_{z,w-v_z}$ [km] | 1.5 | 2 |
| $L_{t,Z_H}$ [h] | 2 | 6 |
| $L_{t,Z_{DR}}$ [h] | 2 | 4 |
| $L_{x,M_S^6}$ [km] | 30 | 45 |

**Table 1.** Summary of the characteristic scales for the two events (see Fig. 2 and 3 for details). $L_{d,X}$ refers to the scale of variation of the variable $X$ in the direction $d$, and $M_S^6$ is the sixth moment of the snow particle size distribution from the WRF simulation, a proxy for $Z_H$.

| | EV1 | EV2 |
|---|---|---|
| Condition 1, $E5 = \frac{\bar{U}}{\bar{W}} \frac{L_{z,Z_H}}{L_{x,M_S^6}} << 1$ | 0.3 | 0.33 |
| Condition 1', $E5 = \frac{\bar{U}}{\bar{W}} \frac{L_{z,Z_{DR}}}{L_{x,Z_{DR}}} << 1$ | 0.375 | 0.55 |
| Conditon 2, $E9 = \frac{L_{z,Z_H}}{L_{z,w-v_z}} << 1$ | 0.4 | 0.2 |
| Conditon 2', $E9 = \frac{L_{z,Z_{DR}}}{L_{z,w-v_z}} << 1$ | 0.33 | 0.15 |
| Condition 3, $E10 = \frac{1}{L_{t,Z_H}} \frac{L_{z,Z_H}}{\bar{W}} << 1$ | $1.0e^{-4}$ | $3.1e^{-5}$ |
| Condition 3', $E10 = \frac{1}{L_{t,Z_H}} \frac{L_{z,Z_H}}{\bar{W}} << 1$ | $8.7e^{-5}$ | $2.3e^{-5}$ |

**Table 2.** Verification of the three analytical conditions derived in Sect. 2 for the two case studies. Conditions 1, 2, 3 correspond to Eq. 6, 8 and 10 evaluated with $Z_H$, while conditions 1', 2', 3' are evaluated with $Z_{DR}$.

The variograms are plotted in Fig. 2 for EV1 and in Fig. 3 for EV2. The obtained values for $\bar{U}$, $\bar{W}$, $L_{t,Z_H}$, $L_{x,U}$, $L_{x,Z}$, $L_{z,Z}$, $L_{x,Z_{DR}}$, $L_{z,Z_{DR}}$ and $L_{z,w}$ are summarized in Table 1.

Table 2 shows that the three environmental conditions derived in Sect. 2 are verified for the bulk of the two case studies for $Z_H$ and $Z_{DR}$. To evaluate an upper limit for condition 1, we use $L_{x,Z_{DR}} \approx 20$ km, because we know $L_{x,Z_{DR}} > 15$ km from the horizontal variogram of $Z_{DR}$ (not shown) but we cannot derive an accurate larger value from WRF simulations (the model hypothesizes spherical particles). We note that the third condition (stationarity) is largely respected while the first and second conditions are less clearly respected. Particular attention should thus be paid to this aspect when analyzing the results in the next section, particularly when focusing on very local (in space and in time) patterns.





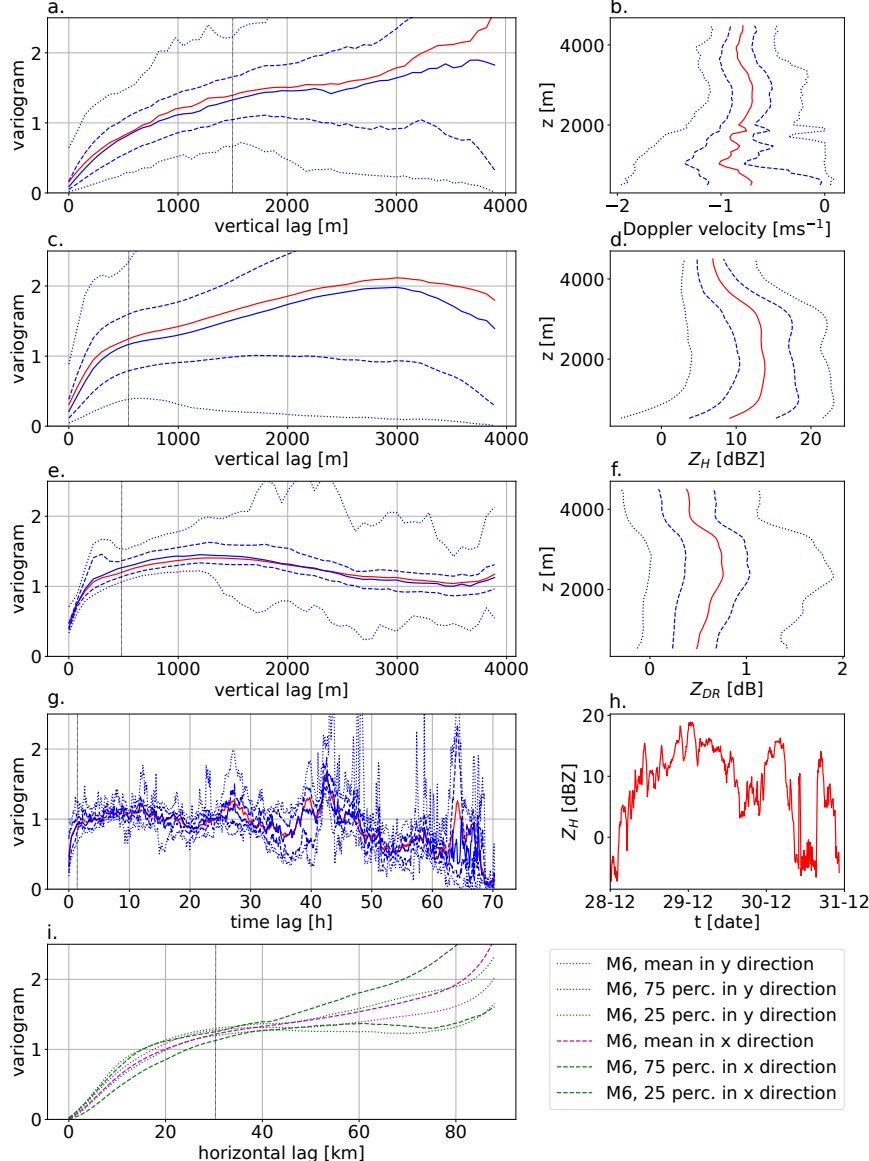

**Figure 2.** Variogram analysis during EV1. (a) vertical empirical variogram of the Doppler velocity along the vertical direction (statistics refer to the temporal dimension). (b) vertical profiles of the Doppler velocity averaged in time. Vertical experimental variograms for $Z_H$ (c) and $Z_{DR}$ (e). Vertical profiles of $Z_H$ (d) and $Z_{DR}$ (f) averaged in time and horizontally. (g) temporal empirical variogram of $Z_H$. (h) spatially averaged $Z_H$ as a function of time. In panels (a) to (h), red lines show the mean profiles, solid blue lines the median profiles, and dashed (resp. dotted) lines the 25-th and 75-th percentiles (resp the 5-th and 95-th percentiles). The vertical black dotted lines in (a), (c), (e), (g) correspond to the characteristic scales measured and reported in Table 1. (i) in dashed (resp. dotted) magenta line, the horizontal variogram in $x$ direction (resp. $y$ direction) of the sixth moment of the snow particle size distribution during all the event from the WRF simulation (see Appendix A). In dashed green (resp .dotted green) line the 25-th and 75-th percentiles in the $x$ (resp. $y$) direction. $x$ and $y$ designate model axes (shifted with respect to longitude/latitude, see Vignon et al. (2019a)).

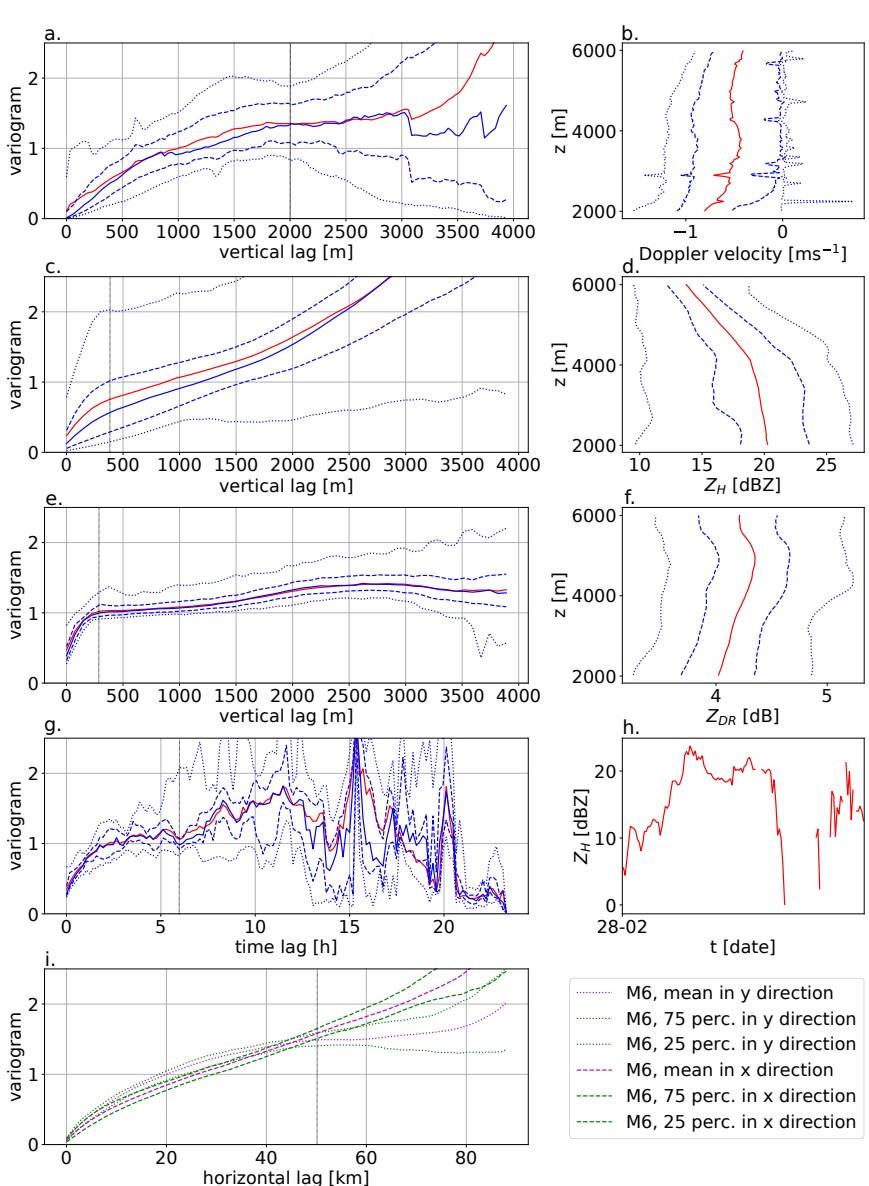

**Figure 3.** Same as Fig. 3 for EV2





| Parameter | EV1 | EV2 |
|:---:|:---:|:---:|
| $p$ [%] | 25 | 25 |
| $D_x$ [km] | 10 | 15 |
| $D_z$ [km] | 4.5 | 6 |
| $\Delta t$ [min] | 10 | 10 |
| $\Delta x$ [km] | 1 | 1 |

**Table 3.** Numerical values of the processing parameters of the PIVS method for each of the two case studies.

Table 3 further presents the processing parameters required for the implementation and selected following the guidelines
of Sect. 3. The sensitivity to their exact values has been assessed (not shown) and we can underline that our results are not
significantly sensitive to a change of these parameters by $\pm 50$ %.

### 4.3 Illustration of the method

Fig. 4 illustrates the process-identification patterns in a RHI scan during EV1. Overall, one can notice that the method reveals
a quite characteristic (and expected) vertical pattern in terms of process occurrence: CG takes place mainly higher than AR
(in this RHI, CG is around $3100 - 4200$ m, AR between $3100$ m and $2000$ m), which is itself higher than SUB (from $2000$
to $500$ m). This pattern is consistent with the typical structure of precipitation at DDU as already described in Grazioli et al.
(2017a); Durán-Alarcón et al. (2019); Vignon et al. (2019a). Cloud ice crystals first grow by vapor deposition, then starts to
sediment, aggregate and eventually rime when supercooled liquid water is present in the clouds. As dry katabatic winds keep
blowing near the ground surface during the event, an approximately 1.5 km-deep layer exhibiting a decrease in reflectivity with
decreasing height is noticeable and reflects the enhanced sublimation of snowflakes.

One should further remember that the conditions to apply the method derived from the theoretical framework in Sect. 2
were verified for the bulk of the event. Subsequently, such conditions may be not respected locally (in time and space). It is
particularly the case for RHIs exhibiting fallstreaks in conditions with strong wind shear as in Fig. 4c-d (identified in blue
in Fig. 4c). Indeed, the reflectivity may increase along the fallstreak while a vertical analysis of the gradient rather reveals a
decrease (and therefore SUB) of $Z_H$ with decreasing height (see the green area in Fig. 4c).

The temporal structure of the process identification by PIVS is presented in Fig. 5 for EV1 and in Fig. 6 for EV2. Interest-
ingly, the stratification of layers that we illustrated for a particular RHI in Fig. 4a-b is clearly visible during most of EV1. SUB
and AR both clearly manifest as well defined layers while CG is the dominant process at the top of the precipitating cloud.
Albeit intriguing, the thin upper layer of SUB visible around 3000 m between 28 December at 14:00 UTC and 29 December
at 09:00 UTC is likely explained by a sub-saturated layer with respect to ice visible in the radiosoundings (not shown, see Fig.
2h from Vignon et al. (2019a)).

Regarding EV2, one can first notice that SUB occurs in the bottom part of the cloud before 04:00 UTC. This corresponds

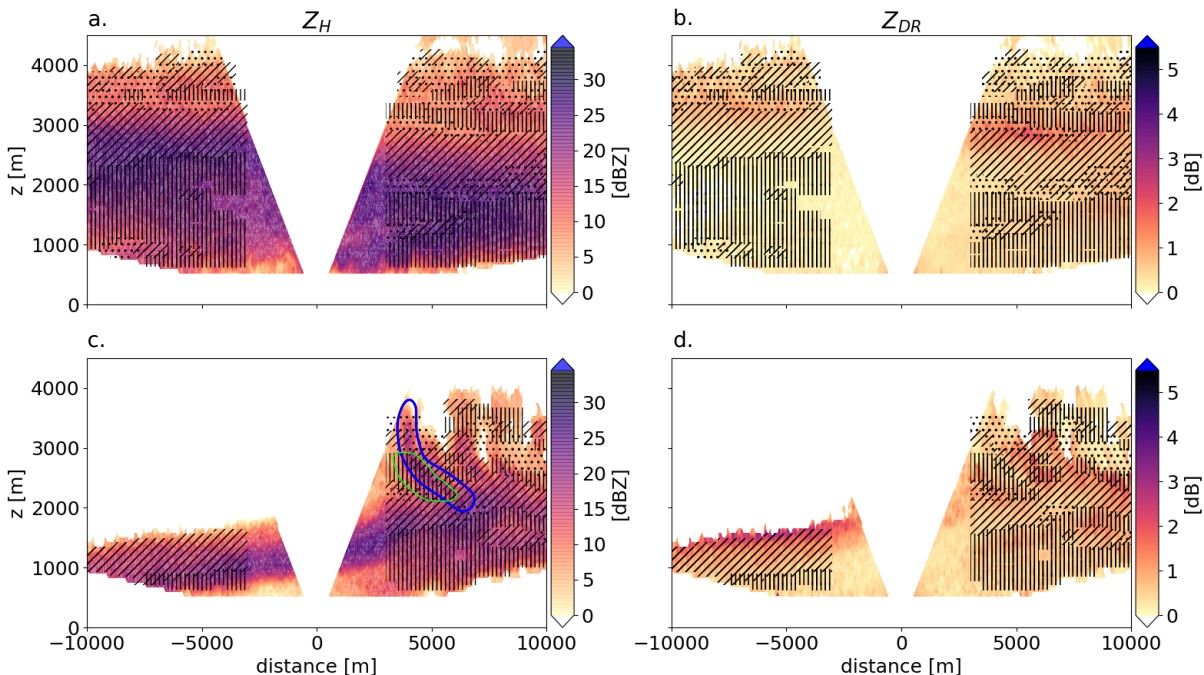

**Figure 4.** Illustration of the PIVS method application on $Z_H$ (a) and $Z_{DR}$ (b) RHI scans during EV1 on December 28, 2015 at 21:48 UTC. (c) and (d): illustration of fall streaks on December 28, 2015 at 10:20 UTC. (c) blue lines highlight fallstreaks and green contours indicate regions where the PIVS method is probably not reliable. In all panels, the process identification of the PIVS method are superimposed with black symbols: dots ∴ locates the CG process, // the AR process and || the SUB process.

to virga clouds ahead of the warm front, which we can identify also in radiosoundings with a very dry layer visible at 00:00

UTC (see Fig. 6b in Gehring et al., 2020b). This is consistent with the conceptual model for precipitation evolution during the passage of the warm front of Gehring et al. (2020b) and previous studies (e.g., Clough and Franks, 1991). Moreover, the PIVS method identifies crystal growth as the dominant process in the top layer of the cloud ($\approx 4500 - 6000$ m) and a layer of aggregation/riming below ($\approx 4500 - 3500$ m) during most of the event (from 06:00 UTC to 12:00 UTC). These two layers persist during 6 hours and are $\approx 500 - 1000$ m deep during the core of the event.

Interestingly, the process identification with PIVS provides a more 'broken-up' pattern during EV2 than during EV1. This concurs with the fast evolving synoptic system associated with the intense warm conveyor belt dynamics over Korea (Gehring et al., 2020b). Notably, a well marked region dominated by sublimation around 08:00 UTC at $\approx 4000$ m can be pointed out. However, the radiosounding at 00:00 UTC reveals supersaturation with respect to ice in the corresponding layer (see Fig. 6b in Gehring et al., 2020b), making snowflakes sublimation impossible. This particular inconsistency can be explained by an

isolated - but strong - turbulent updraft (see more details in Appendix B). As specified in Sect. 3, the PIVS method only holds





in conditions with a negative (towards the ground) net velocity of the hydrometeors and it provides non-realistic results for
very localized - in time and space - profiles with updrafts strong enough to move hydrometeors upward.

### 4.4 Comparison with MASC data and radar-based hydrometeor classification

We now compare the results of PIVS method with surface MASC observations of ice crystals and snowflakes to assess its
robustness. Figures 5f and 6f show time series of the number of hydrometeors of different categories detected by the MASC
during EV1 and EV2 respectively.

During EV1, most hydrometeors are small particles, owing to the intense sublimation by katabatic winds (and possible blow-
ing snow not fully filtered out by the MASC processing, see Section 4.1). Between 00:00 UTC and 05:00 UTC and between
17:00 and 19:00 UTC, 30 December 2015, the MASC detected a significant number of aggregates and graupel particles. Dur-
ing those periods, the PIVS method identifies aggregation/riming as dominant at low levels, together with limited sublimation
near the ground (Fig. 5b-e, red boxes). Moreover, the highest reflectivities measured in the column during these events are
around 00:00 UTC, 29 December, while the MASC detects very few particle. This is consistent with the concomitant enhanced
sublimation identified by PIVS (Fig. 5d, blue box). A few hours later (around 06:00 UTC, orange box), SUB becomes slightly
less active and a larger number of particles is detected by the MASC.


During EV2, the MASC did not detect particles before 04:00 UTC, 28 February (see Fig. 6f) , consistent with the sublimation
layer identified between 2000 and 3000 m a.g.l. by PIVS (Fig. 6d). Particles start being detected by the MASC around 04:00
UTC, which corresponds to the decay of the low-level sublimation layer. The majority of particles identified by the MASC are
aggregates and graupel between 04:00 and 08:00 UTC. However, Gehring et al. (2020b) report a significant difference in terms
of particle size distribution (PSD) over the time period. This explains why the PIVS signal for AR is relatively weak between
04:00 and 06:00 UTC: aggregation is active, but aggregates are small and their polarimetric radar signature is weak. Around
08:00 UTC 28 February 2018, the number of aggregates detected by the MASC reaches its maximum value, and Gehring et al.
(2020b) point an increase in the mean size of observed aggregates. This peak is collocated with a well-defined layer of AR
identified by PIVS above.
We therefore obtain a fair consistency between the type and number of particles identified by the MASC at the surface and the
microphysical processes identified by PIVS aloft.

An additional comparison with hydrometeor proportions from the radar-based hydrometeor classification with demixing
(Besic et al., 2018) provides complementary insights into the reliability and robustness of the PIVS method. One should
nonetheless bear in mind that both the hydrometeor classification and the PIVS method are not completely independent since
they are applied on the same radar data. However, the demixed hydrometeor proportions are determined based on the absolute
values of the polarimetric variables within each individual radar volume (independently of the neighboring ones), while the
PIVS method only relies upon the sign of the local vertical gradients. Importantly, the two methods do not give the same
information. The demixing provides an estimate of the proportion of different hydrometeor types within a radar volume,



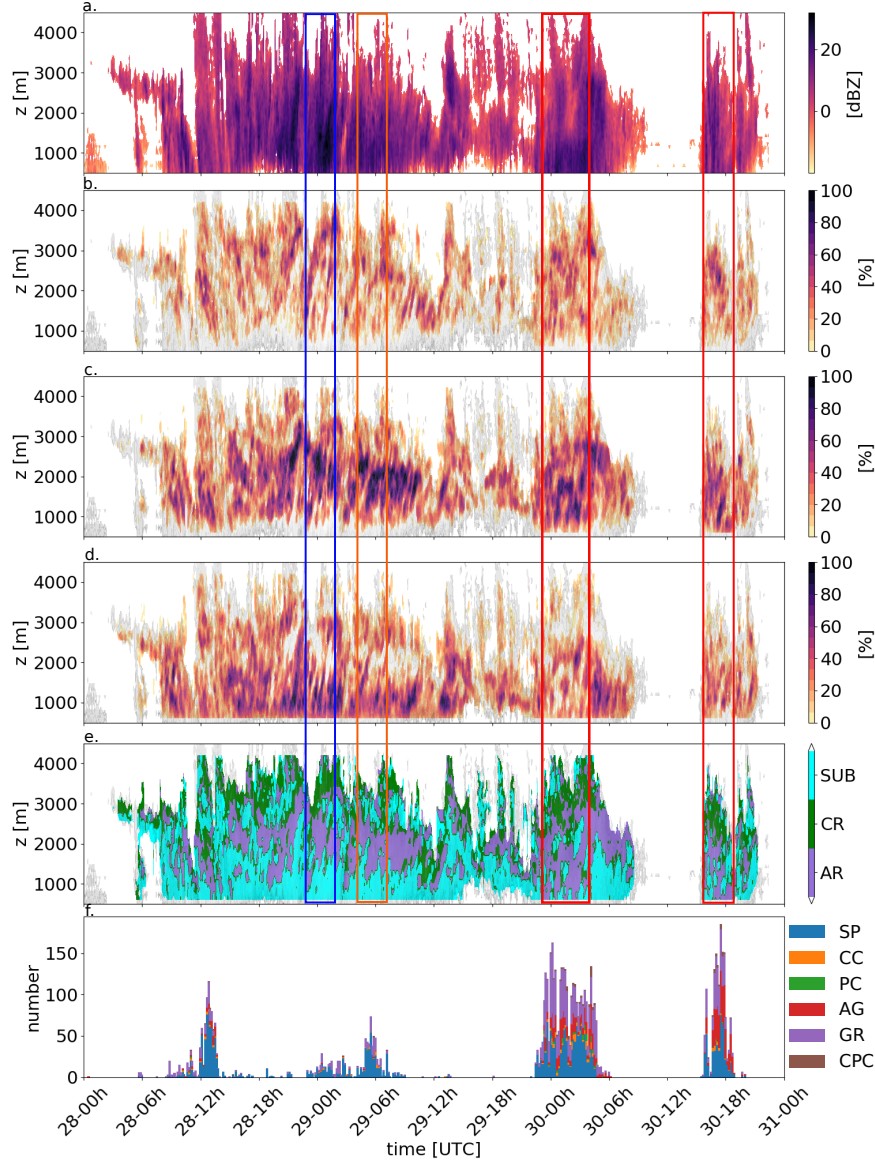

**Figure 5.** Time-height plot of $Z_H$ (a) and of the process identification from the PIVS method during EV1 with (b): CR, (c): AR, (d): SUB and (e): the three categories. In (b)-(e) the gray shading indicates regions where $Z_H > -15$ dBZ. The colorbar in (b)-(d) refers to the percentage of gates at a given height within the RHI corresponding to a certain process. In (e) the color labelling indicates the dominant microphysical process. On the y-axis, z is the altitude above ground level. (f) presents the time series of the cumulative number of hydrometeors - classified into different categories following Praz et al. (2017) - detected by the MASC. SP, CC, PC, AG, GR and CPC refer to small particles, columnar crystals, planar crystals, aggregates, graupel and combination of planar and columnar crystals, respectively. The red boxes delimit time periods during which aggregation is detected down to the ground by PIVS and aggregates and graupel are detected in a relatively high proportion (half of particles) by the MASC. The blue box indicates a period with at the same time intense reflectivity in the column and intense near-surface sublimation associated with few particles at the ground level. The orange box, a few hours later, shows when SUB weakens and when more particles are detected by the MASC.



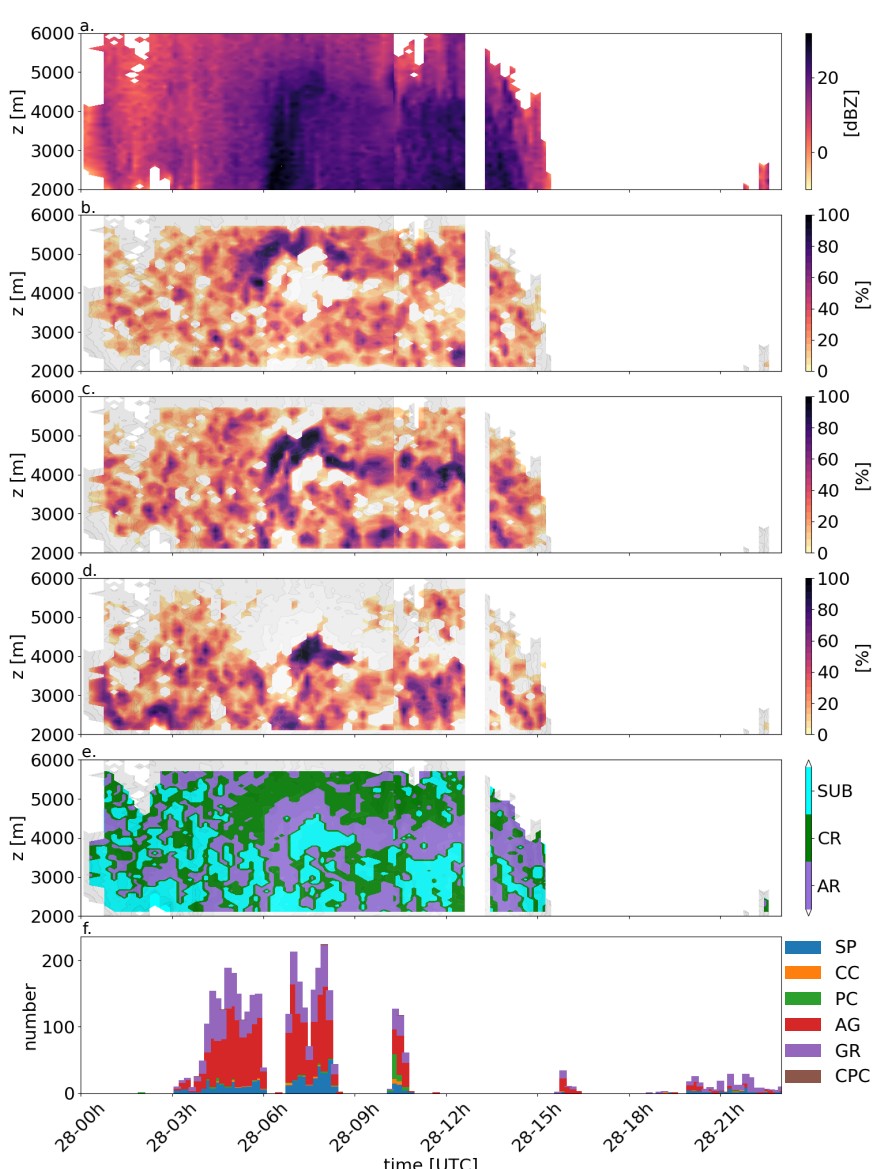

**Figure 6.** Same as Fig. 5 for EV2, without the colored boxes.





whereas PIVS informs about the occurrence of microphysical processes along vertically neighboring radar volumes. The comparison between PIVS output and the demixing products is presented in Fig. 7 for EV1 and in Fig. 8 for EV2. In those two figures, the demixing generally exhibits high crystal (resp. aggregate and rimed particle) concentrations where PIVS identifies CG (resp. AR) as dominant. The difference between the two algorithms is highlighted in Fig. 7c,f and Fig. 8c,f resp, with the plot of the occurrence density for both the demixing and PIVS (see caption for details). The careful inspection of

the averaged vertical distribution during the events (Fig. 7c,f and 8c,f) shows that CG (resp. AR) generally occurs slightly above the maximum concentration in crystals (resp. aggregates and rimed particles). This is consistent with the fact that the concentration of a given type of particles at a given height is mostly determined by the microphysical processes affecting the particles evolution earlier in time (i.e. higher in altitude).

Moreover, one should remember that the radar sensitivity is lower at the top of the signal and the PIVS outcome in the upper

part of the clouds should hence be considered with caution.

Overall, the analysis of the PIVS' products provides relevant insights into the microphysical processes governing snowfall, their occurrence and spatio-temporal organisation. Sect. 4.5 uses this information to provide a statistical characterization of the considered microphysical processes.

**4.5   Microphysical inferences from statistical analyses**

We now gain further insights into the microphysics of the two case studies by carrying out a statistical analysis of the PIVS' output. We use empirical conditional probabilities, defined as the probability $\mathbb{P}$ for a variable $C$ to take the value $c$ conditioned to *process*:

$$\mathbb{P}(C = c | process) = \frac{\mathbb{P}(C = c \cap process)}{\mathbb{P}(process)} = \frac{Counts(C = c \cap process)}{Counts(process)} \tag{11}$$

At this point, it should be noted that PIVS characterizes the dominant processes in terms of radar signature. Because of the strong size dependence, a few large aggregates will dominate quite rapidly the radar signature compared to many more smaller ice crystals. Hence one can expect that aggregation is detected as dominant process while CG persists, and similarly for SUB. This will moderate our analysis in the following section.

**4.5.1   Mean height of process occurrence**

Figure 9a shows the probability distribution of the altitude of occurrence for each process during EV1. SUB is the most frequent process between 500 and 1500 m a.g.l., while CG dominates between 2500 m and 3500 m and AR prevails in between. Note that below 500 m, no data are available and above 4000 m, data might be affected by a decrease in sensitivity of the radar. A clear statistical stratification in terms of dominant processes can be noticed. This stratification is also visible for EV2 (Fig. 10a),

where CG mostly occurs between 4800 m and 6000 m, AR between 4000 and 4800 m and SUB below. One should remember that for EV2, the PIVS method was applied between 2000m and 6000 m a.g.l only.

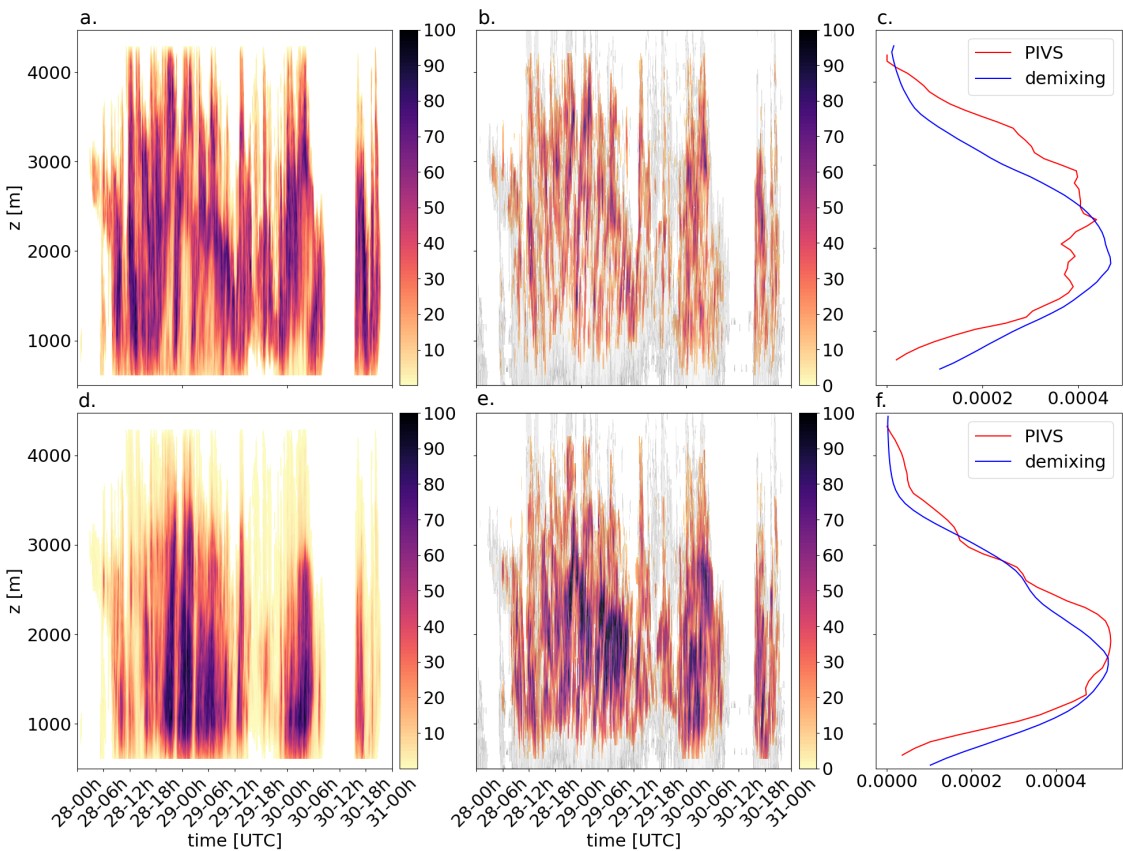

**Figure 7.** (a) (resp. d) shows the time-height plot for EV1 of the proportion of crystals CR (resp. aggregates AG and rimed particles RP) from the demixing algorithm. (b) (resp. e) shows the time-height plot of the proportion of CG (resp. AR) from the PIVS method. (c) (resp. f) shows the vertical plots of the occurrence density of the CR demixed proportion and CG proportion during the event. The proportion for demixing is the number of counts for a specific hydrometeor divided by the total number of counts in this pixel, weighted by the proportion of profiles with significant signal at that height (i.e. with any process identified by PIVS) among all profiles. This allows a meaningful quantitative comparison between the two methods.





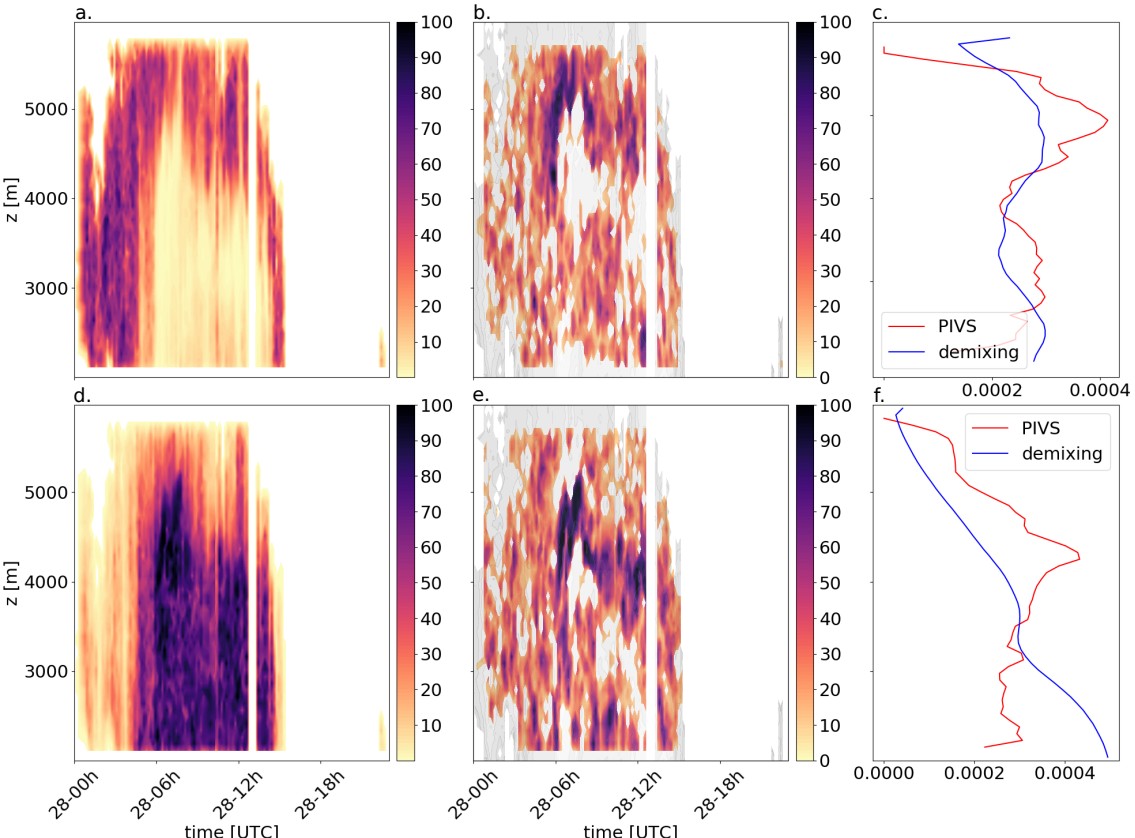

**Figure 8.** Same as Fig. 7 for EV2.

Although the vertical succession of processes is similar between the two events, there is a clear difference in terms of absolute altitude values that can be explained by differences in temperature and humidity profiles driven by the synoptic circulation but also by more local aspects like the dry katabatic layer in Antarctica. The AR layer, however, has a similar thickness in the two

distributions, between 700 m and 1000 m.

### 4.5.2 Magnitude and gradients of $Z_H$ and $Z_{DR}$

Figure 9(b) (resp. c) shows the distribution of the maximal values of $Z_H$ (resp. $Z_{DR}$) associated with a given process, over each vertical profile section during EV1 ($\max Z_H$ and $\max Z_{DR}$). The three processes exhibit different signatures. AR shows the highest mean value for both $\max Z_H$ and $\max Z_{DR}$. Moreover, the right tail of the $\max Z_{DR}$ distribution is the highest for

AR (only slightly for EV2, see Figure 10c) and suggests that the highest values of $Z_{DR}$ - indicating large oblate particles - are mostly identified in AR regions. As AR is defined with positive upward gradients of $Z_{DR}$, this suggests that crystals starting to aggregate or to rime first grow in size, density and oblateness and then progressively become less oblate as they continue to



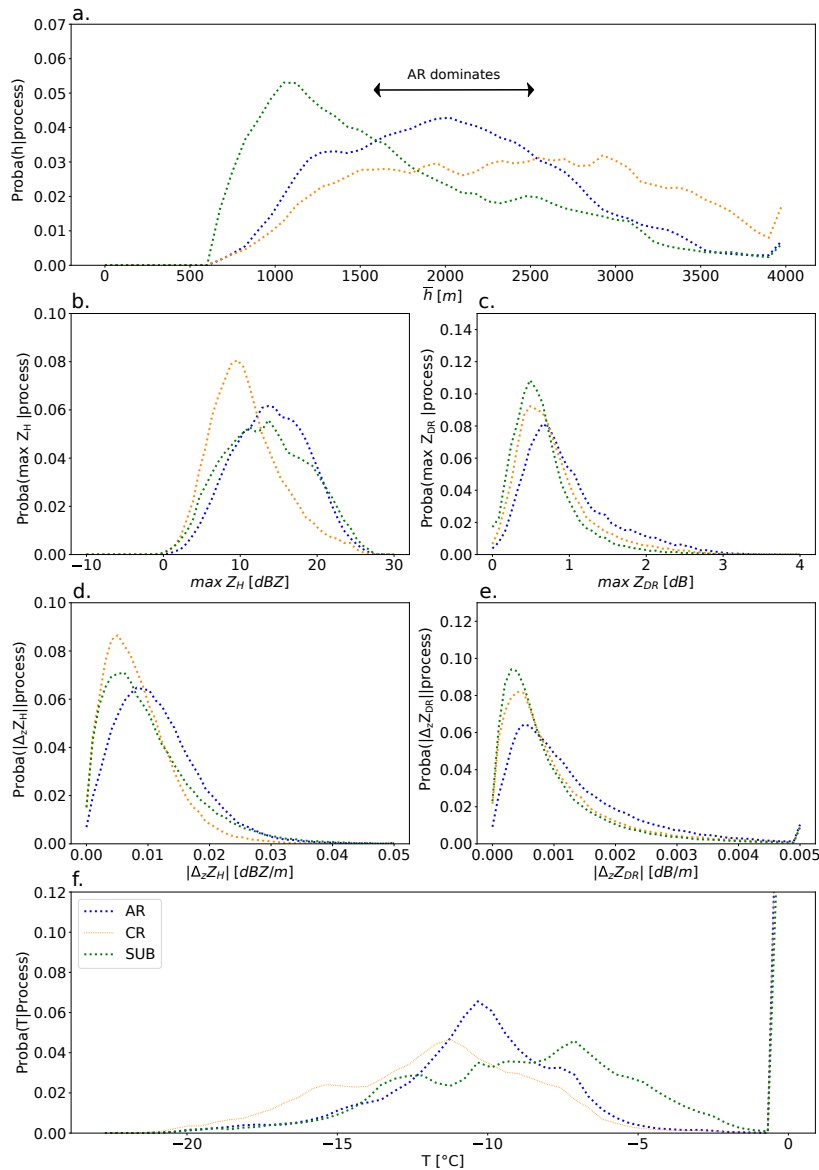

**Figure 9.** Empirical conditional probability distributions for EV1. (a): mean height of process occurrence a.g.l. $\overline{h}$. (b) and (c): absolute maximal value of $Z_H$ and $Z_{DR}$. (d) and (e): absolute value of the mean local vertical gradient of $Z_H$ and $Z_{DR}$. (f): probability of temperature conditioned to different processes. The distributions are computed over all the sections of profiles identified as AR (blue), CG (orange) and SUB (green).

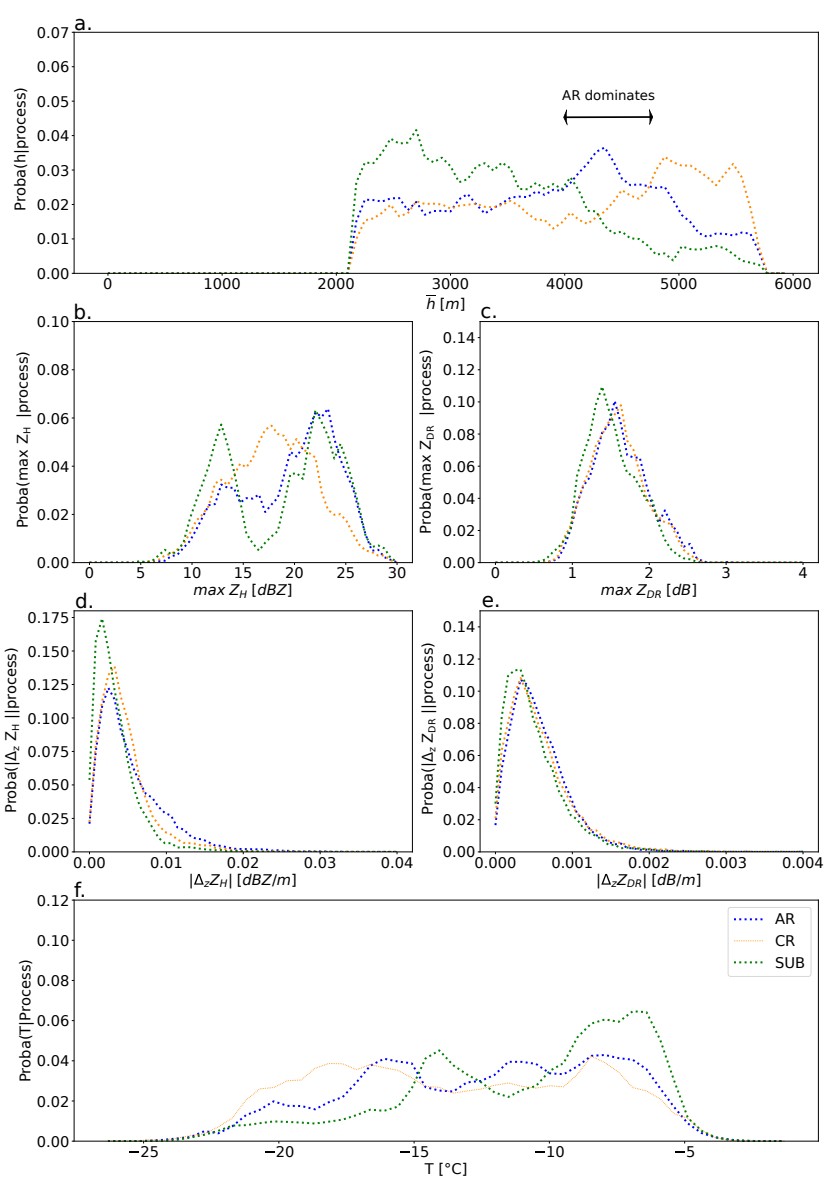

**Figure 10.** Same as Fig. 9 for EV2





grow. This is consistent with the early-aggregates signature proposed by Moisseev et al. (2015).

During EV2, one can distinguish two modes at around 11 dBZ and 20 dBZ in the distribution of $\max Z_H$ for SUB (panel b in
Figure 10). A close analysis of the output of PIVS (not shown) reveals that the first mode corresponds to the sublimation layer
below the warm front during its arrival above the radar (period during which $Z_H$ is relatively weak), whereas the second mode
corresponds to elevated sublimation layers during the core of the event, when the reflectivity reaches higher values.

Figure 9(d). (resp. e) shows the distribution of the local vertical gradient of $Z_H$ (resp. $Z_{DR}$) averaged over the sections
of profiles corresponding to the different processes during EV1 (see Fig. 10d,e for EV2). During the two events AR exhibits
the largest $|\Delta_z Z_H|$ and $|\Delta_z Z_{DR}|$ means but also the highest tails for $|\Delta_z Z_H|$. $|\Delta_z Z_H|$ for SUB exhibits different statistical
signatures between EV1 and EV2: the right tail of the distribution is the shortest among the three categories of processes during
EV2 but comparable to that of AR and larger to that of CR during EV1. We believe this is explained by the strong katabatic
winds during EV1 which are responsible for an enhanced sublimation. AR exhibits the largest gradients of both $Z_H$ and $Z_{DR}$.
Because AR and CG are both associated with negative upward relative gradients of $Z_H$, it means that on average AR increases
more the particle size and/or density than CG. This result concurs with the fact that aggregation and riming are more efficient
processes to increase the size of snowflakes than vapor deposition (Ryzhkov and Zrnić, 2019).

### 4.5.3 Temperature

Temperature is a key parameter influencing the occurrence of microphysical processes. It was for instance identified as a driv-
ing factor for aggregation efficiency (Hobbs et al., 1974; Connolly et al., 2012). Hobbs et al. (1974) distinguish two different
preferential modes for aggregation at two temperatures: one associated with mechanical aggregation (dendrites that mechani-
cally intricate with each other) and one with 'sticking' aggregation at warmer temperatures (crystals that bound together thanks
to their sticky melting surface).

Figures 9f and 10f present the normalized distribution of temperature knowing a process for the two events. Temperature
is obtained from WRF simulation, see Appendix A for details. For AR, EV2 presents two peaks, around $-8°$C and $-17°$C
while EV1 only shows a very well defined peak at $-10°$C. We suggest that the second peak for EV2 (coldest) and the peak
for EV1 may correspond to the mechanical entanglement of aggregation described by Hobbs et al. (1974) and Connolly et al.
(2012) (with a slight difference in temperature: Hobbs et al. (1974) suggests a temperature range between $-10°$C and $-15°$C
and Connolly et al. (2012) observes a maximum in aggregation efficiency at $-15°$C). However, riming cannot be disentangled
from aggregation in AR, and possibly influences these signatures. CG has a wider temperature distribution ranging from
$-7°$C to $-17°$C during EV1 and from $-5°$C to $-21°$C during EV2. This is consistent with the theoretical models for vapor
deposition with different growth habits depending on supersaturation with respect to ice and with temperature down to $-38°$C
(e.g., Nakaya, 1954; Libbrecht, 2005). The two signatures of SUB for both events are similar: they exhibit a maximum around
$-6°$C and a secondary mode at colder temperatures ($-13°$C and $-14°$ for EV1 and EV2 resp.). The secondary mode is less
accentuated for EV1, probably due to the katabatic layer that is responsible for an intense and continuous low-level sublimation





above the Antarctic coast. The high temperature range thus corresponds to a low-level very dry air where sublimation is the dominant microphysical process.

## 5 Conclusions

This study presents the development and application of a new method named PIVS to automatically detect the occurrence of microphysical processes controlling snowfall growth and evolution from dual-polarization Doppler radar scans. PIVS is based on the analysis of the sign of the vertical gradients of $Z_H$ and $Z_{DR}$ extracted from RHI scans. Three classes of microphysical processes (aggregation and/or riming, crystal growth by vapor deposition and sublimation) are identified by jointly observing the sign of the gradients along vertical profiles of the two variables. PIVS differs from hydrometeor classification methods that rely on the absolute value of polarimetric variables. It rather focuses on the vertical evolution of the characteristics of the particles with microphysical processes. In addition it is insensible to calibration errors that can sometimes be an issue when using the $Z_{DR}$ quantity.

The environmental conditions in which the local vertical gradients of polarimetric signal primarily reflects the microphysical processes are first theoretically established. We derive three analytical conditions depending on the typical scales of spatio-temporal variations of the main variables shaping the radar signal ($Z_H$, $Z_{DR}$, $\boldsymbol{u}$, $v_z$). These conditions ensure that (i) the horizontal transport of $Z_H$ and $Z_{DR}$ is negligible with respect to its vertical evolution, (ii) the vertical divergence of $(w - v_z)Z_H$ is mainly driven by the variation of $Z_H$ itself and not by variations in relative vertical velocity and (iii) the timescales of variations of $Z_H$, $Z_{DR}$ due to microphysical processes are sufficiently large so that the profiles of $Z_H$ and $Z_{DR}$ can reach a local quasi-equilibrium with the microphysical processes. These conditions, verified at the scale of the entire event, are sufficient conditions to reliably use PIVS.

We apply and illustrate PIVS on two frontal snowfall cases: one at Dumont d'Urville, Adélie Land, Antarctica and one in the Taebaeck mountains, South Korea. The robustness of the method is assessed by successfully comparing the output with two complementary datasets: snowflake observations from a MASC at the ground and products from a hydrometeor classification based on polarimetric radar data. This comparison highlights local limits of the method, especially in strong updrafts or fallstreaks i.e. when the three conditions, albeit respected over the bulk of the event, do not hold locally. We then use PIVS output to better understand and characterize the microphysical processes at play during these two events. In particular, we could establish the mean altitude and the vertical extent of each process. In Antarctica, aggregation and riming are dominant over a layer extending from 1500 m to 2500 m a.g.l., with crystal growth by vapor deposition above and sublimation below owing to the very dry low-level katabatic layer. In South Korea, the layer of aggregation and riming is slightly thinner and higher, located between 4000 m and 4800 m a.g.l.. PIVS also allows us to characterise the statistics of radar variables ($\max Z_H$, $\max Z_{DR}$, $|\Delta_z Z_H|$ and $|\Delta_z Z_{DR}|$) conditioned to the microphysical processes. Potential early aggregate signatures have for instance been identified and a complementary analysis of the processes' occurrence dependence to temperature shows signatures that may be



associated to the onset of mechanical aggregation.


The present analysis could be easily replicated to other events in polar or mountainous environments sampled by a polarimetric radar, making it possible to better understand the processes governing the formation and evolution of snowfall in different meteorological contexts. Future methodological developments may include additional polarimetric variables, like $K_{dp}$, to help identify and distinguish additional processes (in particular secondary ice generation processes, Ryzhkov and Zrnić, 2019 or

early aggregate generation, Moisseev et al., 2015). Finally, preliminary works not shown here have suggested that PIVS'output can be useful to evaluate the ability of atmospheric models to reproduce the snowfall microphysical processes at the correct location and time. Such an application of the PIVS method would deserve further exploration in the future.

*Data availability.* The APRES3 campaign data are freely distributed on the PANGAEA data repository (https://doi.org/10.1594/PANGAEA. 883562, Genthon et al., 2018). Further information on the ICE-POP dataset and its distribution is given in Gehring et al. (2020a).

**Appendix A: Numerical simulations with WRF**

To get access to the high-resolution spatio-temporal temperature field around the measurement sites and to complement the scale analysis in Sect. 4.2, we run numerical simulations with the 4.0 version of the Weather Research and Forecasting (WRF) model. The simulations set-up is similar to the one presented in Vignon et al. (2019a). It consists in a downscaling method where a 27-km resolution parent domain contains a 9-km resolution nest, which contains a 3-km resolution domain, which itself

contains a $102 \times 102$ km$^2$ nest centered over the measurement site (either DDU or GWU) at a 1-km resolution. To allow for a concomitant comparison with observations and to ensure realistic synoptic dynamics in the model, the wind field in the 27 and 9 km-resolution domains has been nudged towards ERA5 reanalysis (Hersbach et al., 2020). The physical package is similar to the one of Vignon et al. (2019a). In particular the 2-moment (for rain, ice, snow and graupel categories) microphysical scheme from Morrison et al. (2005) is employed. As the snow particle size distribution is assumed to have an exponential shape, the

$k^{th}$ moment of the distribution $M_S^k$:

$$M_S^k = \int\limits_0^\infty D^k N_s^0 e^{-\lambda_S D} dD \qquad (A1)$$

can be calculated from the standard output of WRF i.e. the snow mass mixing ratio $Q_S \left[ \text{kg} \cdot \text{kg}^{-1} \right]$ and the number concentration $N_S \left[ \text{kg}^{-1} \right]$ as follows:

$$M_S^k = \rho \frac{\Gamma(k+1)}{\Gamma(4)^{k/3} Q_S^{k/3} N_S^{1-k/3}} \qquad (A2)$$

In those equations, $\rho$ is the air density, $D$ is the particle dimension and $N_S^0$ (resp. $\lambda_S$) is the intercept (resp. slope parameter) of the size distribution. Otherwise mentioned, the analysed model fields are those from the 1-km resolution domain.





**Appendix B: Local unreliability of the method in case of strong turbulent updrafts**

In strong turbulent updrafts, it can happen that the velocity of the particles becomes positive. In such conditions, as explained in Sect. 3.2, the criteria for identifying a microphysical process based on the sign of the vertical gradients of $Z_H$ and $Z_{DR}$
should be inverted.

In our dataset due to the instrumental configuration, we cannot retrieve the vertical Doppler velocity jointly with $Z_H$ and $Z_{DR}$. During EV1, vertical velocity measurements are available strictly above the radar, in a region where we cannot estimate $Z_{DR}$. During EV2, the cloud profiler WProf - which provides frequent Doppler velocity profiles - is located at MHS, 20 km away from MXPol. Hence we cannot precisely detect the regions where the vertical Doppler velocity is positive within RHIs scans.
By inspecting the time evolution of the Doppler velocities from the vertical profiles, we can pinpoint time periods during which we can expect the application of PIVS not to be reliable. On one hand, EV1 exhibits few updrafts visible in the vertical Doppler velocity height-time plots (see A2). These sparse updrafts are mainly located in the katabatic layer, which is more turbulent than the upper layers (e.g. Denby, 1999; Vignon et al., 2020) but remain limited in amplitude and temporal extent. On the other hand during EV2, strong updrafts are observed between 07:00 UTC and 09:00 UTC (see A1 and Gehring et al.,
2020b). These updrafts last longer than during EV1 and their vertical extent reaches almost 2000 m. The atmosphere above the Taebeck mountains - and thus probably the one above MXPol too - therefore experience a very unstable period questioning the reliability of the PIVS method at this specific time.

Setting a radar profiler up *within* the RHI scan could help accurately locate the turbulent and unstable regions and could
provide interesting information about Doppler velocity characteristics for the three processes.





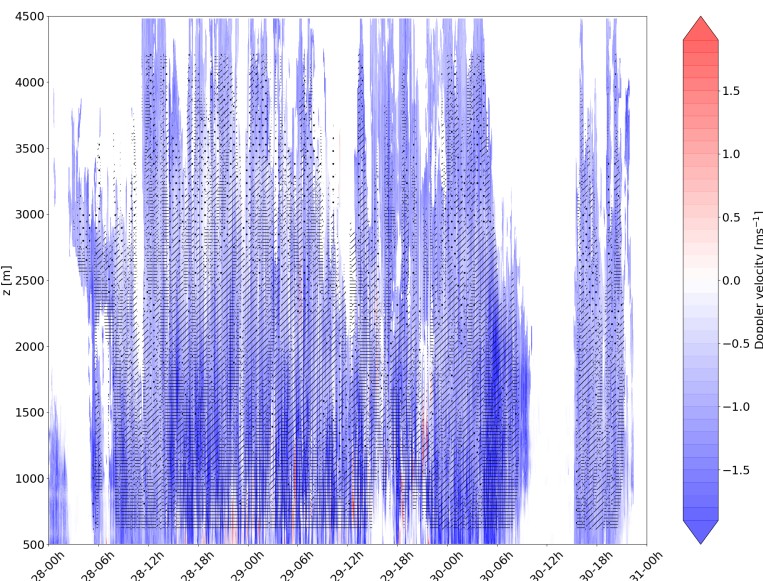

**Figure A1.** Vertical Doppler velocity time height plot during EV2 (WProf data). We superimpose the PIVS output, dots ∴ stand for CG, // stand for AR and ≡ stand for SUB.

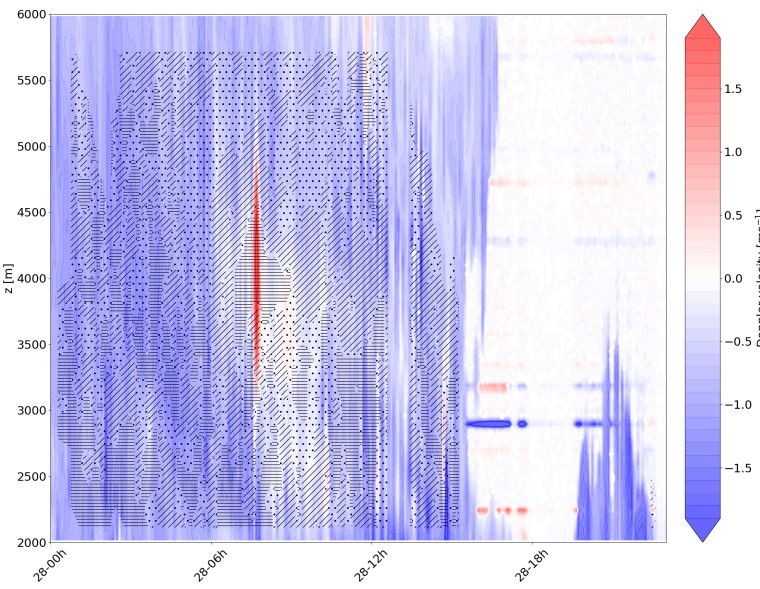

**Figure A2.** Same as A1 for EV1 with MXPol Doppler measurments



*Author contributions.* NP, JG, EV and AB designed and conducted the study. NP carried out the major part of the analysis. NP and JG processed the radar data. NP and EV derived the theoretical framework. EV ran the WRF simulations. NP prepared the manuscript with contributions from all authors.

*Competing interests.* The authors declare that they have no conflict of interest.

*Acknowledgements.* JG acknowledge the financial support from the Swiss National Science Foundation (grants 175700/1). EV's contribution has been financed by the EPFL-LOSUMEA project. We gratefully thank Nikola Besic for running the demixing algorithm over the two case studies. We are are greatly appreciative to the participants of the World Weather Research Programme Research Develop-ment Project and Forecast Demonstration Project, International Collaborative Experiments for Pyeongchang 2018 Olympic and Paralympic15winter games (ICE-POP 2018), hosted by the Korea Meteorological Administration. We also acknowledge the support of the French National Research
Agency (ANR) and of the French Polar Institute (IPEV) to the APRES3 project (http://apres3.osug.fr) .





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
