# Peer review of "Identification of snowfall microphysical processes from Eulerian vertical gradients of polarimetric radar variables"

_Atmospheric Measurement Techniques, 2020_

## Referee Comment (RC1) · Anonymous Referee #1 · 30 Dec 2020

The authors of the manuscript suggest to use the signs of the vertical gradients of Z and ZDR to identify three key microphysical processes of snow formation: depositional growth, aggregation, and sublimation. Their intent is to recognize particular processes rather than to perform hydrometeor classification which was in the focus of various studies during last two decades. Such motivation is great but I am afraid that the methodology is too simplistic to reflect the complexity of ice / snow formation in real clouds. I am particularly concerned about the notion that any positive vertical gradient of Z is a manifestation of sublimation. Nonmonotonic vertical profiles of Z in ice parts of the clouds are very common. For example, they can be attributed to a "pulse nature" of the ice formation near the tops of the clouds when different pockets of ice generated

aloft sediment one after another producing some periodicity in the vertical profile of Z. Deposition occurs due to supersaturation with respect to ice, and sublimation takes place where air is undersaturated with respect to ice. Therefore, these processes are dictated by the spatial distribution of humidity (as well as temperature) within the clouds. Unfortunately, the authors do not connect the results of radar identification with the thermodynamic structure of the atmosphere in the two cases examined. Referring to the papers of Vignon or Gehring et al. is not sufficient. My suggestion is to modify Figs. 5 and 6 by overlaying the temperature isotherms and a contour of the RHi = 100%, so that the reader will be able to check consistency between the classification results and atmosphere stratification. I also recommend to add similar figures for ZDR. My another concern is that the computation of the gradients is performed along the vertical rather than along the Lagrangian fall trajectories of snowflakes. The paper is overburdened with a secondary stuff which may not be very relevant to the primary idea of the study such as Tables 1 and 2 and Figs. 2 and 3. It may not be necessary to specify the altitudes at which CG, AR, and SUB dominate for the two analyzed events in the abstract and conclusions. These are not typical height intervals where the three processes predominantly occur in the Antarctic region or Korea and this is not a climatological study where such generalization is appropriate. Lines 190 – 195. A primary reason for ZDR to decrease with aggregation is the reduction of the ensemble density of snowflakes (because larger snowflakes have lower density) and more chaotic orientation of aggregated snowflakes. Change of the particles shape (if any) plays a secondary role. Lines 435 – 440. The density of aggregated snowflakes is inversely proportional to their size. Therefore, the ensemble density of snowflakes decreases in the course of aggregation.
* * *

---

## Referee Comment (RC2) · Anonymous Referee #2 · 18 Feb 2021

Identification of snowfall microphysical processes from vertical gradients of polarimetric radar variables

By Planat et al.

Identification of snowfall microphysical processes using radar observations is a challenging topic, because of the similarity of the radar signatures of some of the processes. Changes of radar variables as a function of altitude carry additional information. The authors are proposing a new method that can be used to advance our interpretation of radar observations of ice microphysical processes.

The manuscript is structured well and clear. I have listed a few comments that I would

like the authors to address before the paper can be accepted.

General comments: - I really like how the authors used the continuity equation to derive conditions which could be used to diagnose if vertical gradients of radar variables are driven by microphysics. Because this part is rather new, for the radar community, it would be good to explain in a few sentences what these conditions actually mean. For example, the condition 1 indicates that the reflectivity field is horizontally homogeneous.

Related to the continuity equation, have you checked if the conditions 1,2, and 3 are the same if you are using reflectivity and differential reflectivity. I wonder if the underlying sensitivity of these radar variables is different.

- Case studies. I miss a figure showing Z and Zdr HTI figures for the case studies. I wonder if they can be included, as a supplementary material for example.

- Sublimation. This is probably the most challenging class for my understanding and interpretation. In Fig. 5 and 6 you show the PIVS results. The SUB class is rather prevalent and sometimes its appearance is puzzling. For example, in Fig. 5 starting from the cloud top, PIVS shows CR then SUB then AR then SUB. I cannot figure out how the transition from CR to SUB and then to AR could happen. It would be very helpful to see radio sounding measurements and more discussion on what SUB actually means in this case.

Specific comments:

Lines 193-194: "as particles become more spherical (less oblate)" this is an incomplete statement, also changes in particle density would affect Zdr.

Lines 198-200: "Crystal growth by vapor deposition (hereafter CG) corresponds to an increase in particle size with decreasing height ($\partial z$ ZH < 0) and an increase in oblateness with decreasing height ($\partial z$ ZDR < 0) as particles generally grow along their longest dimension (Schneebeli et al., 2013; Andric ÌĄ et al., 2013; Grazioli et al., 2015)"

In some cases, ice crystal growth would also change density. For example, growing dendrites will become more oblate, but at the same time the density would decrease. These two processes affect dual -polarization radar signatures in opposite direction. I wonder if the statement that ZDR will increase as particle grow by vapour deposition is valid generally? Could you clarify this point.

---

## Author Comment (AC1) · 9 Apr 2021

Please find in this document our answers to both referees' comments. We thank the two anonymous reviewers for their useful comments that helped improve the manuscript. We hope that our corrections to the manuscript will make it suitable for publication in AMT.

Please also note the supplement to this comment:
https://amt.copernicus.org/preprints/amt-2020-463/amt-2020-463-AC1-supplement.pdf

---

## Author Response (AR1)

**Response to reviews, AMT Manuscript amt-2020-463**
**Identification of snowfall microphysical processes from Eulerian vertical gradients of polarimetric radar variables**

*Dear AMT Editor,*
*Please find in this document our answers to the referees' comments. We thank the two anonymous reviewers for their useful comments that helped improve the manuscript. We hope that our corrections to the manuscript will make it suitable for publication in AMT.*
*Yours sincerely,*
*The authors of "Identification of snowfall microphysical processes from Eulerian vertical gradients of polarimetric radar variables"*

**Referee 1**

The authors of the manuscript suggest to use the signs of the vertical gradients of Z and ZDR to identify three key microphysical processes of snow formation: depositional growth, aggregation, and sublimation. Their intent is to recognize particular processes rather than to perform hydrometeor classification which was in the focus of various studies during last two decades.

Such motivation is great but I am afraid that the methodology is too simplistic to reflect the complexity of ice / snow formation in real clouds.

We gratefully thank Reviewer #1 for his/her review that helped us improve our paper. We hope our answers to his/her comments will meet his/her expectations and correct and clarify some aspects of the paper.

We aimed to develop a simple method at first, based on two polarimetric variables only, to illustrate the potential of vertical gradients to identify microphysical processes in conditions when the microphysics drives the vertical variability of the radar signal. Further developments of the method will include other polarimetric variables ($K_{dp}$ and $\rho_{hv}$) to make the characterization of processes more accurate. This paper presents the development of the method and its potential, but a second important objective is to assess in which conditions (meteorological situation, data analysis methodology) the complexity of the microphysical processes at play can be deciphered from the analysis of the vertical structure of radar variables. We hope that the revised version of the manuscript - in which we have taken into account the reviewer's comments – make clearer that we are aware of the complexity of ice/snow formation in clouds and that our methodology has been in fact developed to extract information from this complex system.

1) I am particularly concerned about the notion that any positive vertical gradient of Z is a manifestation of sublimation. Nonmonotonic vertical profiles of Z in ice parts of the clouds are very common. For example, they can be attributed to a "pulse nature" of the ice formation near the tops of the clouds when different pockets of ice generated aloft sediment one after another producing some periodicity in the vertical profile of Z. Deposition occurs due to supersaturation with respect to ice, and sublimation takes place where air is undersaturated with respect to ice. Therefore, these processes are dictated by the spatial distribution of humidity (as well as temperature) within the clouds.

We fully agree with this comment, which is somehow related to the issue of fallstreaks that we briefly discuss in the paper. Fallstreaks produce vertical gradients of radar variables that are not due to the microphysical evolution of snowflakes during their fall but reflect horizontal heterogeneities of ice crystals' and snowflakes' generation and advection. These patterns motivated our theoretical analysis (see Sect. 2) in which we derive sufficient meteorological conditions for the vertical gradients to give access to reliable microphysical information. These sufficient conditions are based on a discussion on the relative characteristic scales of the inhomogeneities in the radar fields ($Z_H$, $Z_{DR}$) in both the vertical and the horizontal directions, as well as temporally. When met, these conditions prevent the signal from being filled with patterns such as fallstreaks or ice pockets at the scale of the event and if present locally (spatially and temporally), such patterns nonetheless do not dominate the vertical signal and its analysis.

To clarify this point in the manuscript, we modified lines 80-92 introducing the theoretical part (see below). We address more specifically the links with the thermodynamics fields (relative humidity temperature) in our answers to questions 2) and to question 4) from Reviewer #2.

"One may nonetheless question the approach when additional mechanisms alter the evolution of radar variables along the vertical direction. This is for example the case when strong generating cells are present at the top of the cloud and the associated precipitating particles sediment. When

advected, the horizontally heterogeneous snowfall manifests as so called 'fallstreaks' which induce vertical gradients of polarimetric variables. Analyses of microphysical processes along fallstreaks (i.e. following the snow particles in a Lagrangian framework) are very relevant and give insights into the microphysical evolution of complex precipitation systems (e.g., Pfitzenmaier et al. 2018). However, fallstreak retrieval algorithms are based on the accurate acquisition of the 3D wind field which is often not available from measurements. Since the proposed method of snowfall microphysical process characterization will be based on the interpretation of local vertical gradient in Eulerian vertical profiles of polarimetric radar variables (see Sect. 3), it is hence crucial to clearly define the meteorological conditions in which such gradients give access to reliable information about microphysical processes and therefore the conditions in which our method can be applied. These conditions will be dictated by both the 3D wind field (to avoid e.g. strong updrafts conditions or significant wind shear favorable for fallstreaks) and the spatial heterogeneity of the radar fields $Z_H$ and $Z_{DR}$) "

Finally, these advection-induced vertical gradients are illustrated in Fig. 2 and commented at lines 341-345 and in the Appendix B. We are fully aware that they limit and constrain the application of our method but we argue their impact is limited when the results are interpreted statistically.

2) Unfortunately, the authors do not connect the results of radar identification with the thermodynamic structure of the atmosphere in the two cases examined. Referring to the papers of Vignon or Gehring et al. is not sufficient. My suggestion is to modify Figs. 5 and 6 by overlaying the temperature isotherms and a contour of the RHi = 100%, so that the reader will be able to check consistency between the classification results and atmosphere stratification.

We understand the concern of the reviewer and the importance of connecting the radar observations and PIVS output to the thermodynamic fields. Following this recommendation, we have updated Figs. 5 and 6 (now Figs. 3 and 4) with the radiosoundings available for each event. The RS are launched at a daily frequency during EV1 and every 6 hours during EV2. We have also added the temperature field (from WRF simulation, see appendix A) in Figs. 3a and 4a. The radiosoundings are used in the manuscript to discuss PIVS output with in particular the SUB layer at 3000m during EV1 that we believe is related to the dry layer visible in the corresponding sounding.
However, a closer comparison between the relative humidity field from the model simulation and PIVS output is questionable, as the model fields above the stations can be biased or slightly shifted in space and time. This last point is illustrated in the figures below, where we plot the radiosoundings together with the model vertical profile (extracted along the RHI at the closest timestep and averaged). In particular, during EV2 in Korea, the model predicts $RHi > 100$ % during most of the event, but this in not measured by the RS. Similarly, the very dry near-surface layer observed on December 28 during EV1 is only partially represented by the model. Therefore, we argue that a direct comparison between PIVS outputs and the relative humidity fields form WRF would not provide very robust information.

[Figure]

Comparison of the RS with WRF outputs for the temperature (left) and RHi (right) during EV2.

[Figure]

Comparison of the RS with WRF outputs for the temperature (left) and RHi (right) during EV1.

[Figure]

New Fig 3 of the manuscript (before Fig. 5) with the addition of the RS (g)-(j), temperature field (a) and the precipitation rate in (f).

[Figure]

New Fig. 4 of the manuscript (before Fig. 6) with the addition of the RS (g)-(j), temperature field (a) and the precipitation rate in (f). Grey shaded area in (f) indicates periods during which the MASC was not working.

[Figure]

Supplementary Fig. S3: Height-time plot of $Z_H$ (a) and $Z_{DR}$ (b) for EV1, together with the dominant process type identified with the PIVS method (c). Dashed green (a,b) and black (c) lines show the temperature obtained from the WRF numerical simulation (see Appendix A of the main manuscript for details) and extracted along the same RHI as the radar. For measured temperature and RHi, please refer to the Radiosoundings visible on Fig.3 of the paper

[Figure]

Supplementary Fig. S4: Same as S3 for EV2.

3) I also recommend to add similar figures for ZDR.

We have added two figures in supplementary material with Z_H, Z_DR and PIVS output together with the temperature field (Figs. S3 and S4, see above).

4) My another concern is that the computation of the gradients is performed along the vertical rather than along the Lagrangian fall trajectories of snowflakes.

Indeed, we fully agree that ideally, the characterization of the microphysical evolution of snowflakes should be performed in a Lagrangian framework. However, the retrieval of Lagrangian trajectories requires the frequent sampling of the 3D wind fields. Such measurements are generally not available, and one is often constrained to work in an Eulerian framework (e.g. Ryzhkov et al. (2016), Andric et al., 2013, Tiira and Moisseev (2020)). This in fact motivated the development of our method, as well as our theoretical derivations based on the continuity equation to assess in which meteorological conditions a vertical (Eulerian) analysis of the snowfall can reliably provide information about the underlying microphysics.
To clarify this point in the paper, we modified the title ('vertical gradients' to 'Eulerian vertical gradients') and modified the introduction of the theoretical derivation (see the answer to question 1).

5) The paper is overburdened with a secondary stuff which may not be very relevant to the primary idea of the study such as Tables 1 and 2 and Figs. 2 and 3.

Following this comment, Figs. 2 and 3 have been moved to supplementary material. However, Tables 1 and 2 are necessary to assess the applicability of the method on both case studies. Both Tables have therefore been kept in the main manuscript, and we have modified the comments on Table 2 to specify its role:
"Table 2 shows that the three environmental conditions derived in Sect. 2 are verified for the bulk of the two case studies for ZH and ZDR, this therefore legitimizes the use of vertical gradients and the applicability of PIVS."
Table 3 (method's parameters) has also been moved to the supplementary material.

6) It may not be necessary to specify the altitudes at which CG, AR, and SUB dominate for the two analyzed events in the abstract and conclusions. These are not typical height intervals where the three processes predominantly occur in the Antarctic region or Korea and this is not a climatological study where such generalization is appropriate.

We thank the reviewer for this comment. We have reformulated the end of the abstract as follow:

"In particular, we are able to automatically derive and discuss the altitude and thickness of the layers where each process prevails., for each case study."
We also have modified the conclusion following this recommendation, we now summarize the microphysical observations with "We derive characteristic metrics of the microphysical processes, in particular the altitude of layers at which each process is dominant, and the vertical extension thereof."

7) Lines 190 – 195. A primary reason for ZDR to decrease with aggregation is the reduction of the ensemble density of snowflakes (because larger snowflakes have lower density) and more chaotic orientation of aggregated snowflakes. Change of the particles shape (if any) plays a secondary role.

We thank the reviewer for this correction. We modified the manuscript lines 204-207 with the following:
"Aggregation and riming (hereafter AR) correspond to an increase in reflectivity due to an increase in particle size and/or density with decreasing altitude ($\partial_z Z_H < 0$) and a decrease with decreasing height in $Z_{DR}$ ($\partial_z Z_{DR} > 0$). This decrease in $Z_{DR}$ is due to the decrease in particle density (larger snowflakes being less dense) and to the more chaotic orientation of the snowflakes associated with their increase in size and Reynolds number (Li et al., 2018; Ryzhkov and Zrnic, 2019)."

8) Lines 435 – 440. The density of aggregated snowflakes is inversely proportional to their size. Therefore, the ensemble density of snowflakes decreases in the course of aggregation.

Thank you for this correction. We modified the corresponding lines (456-457):

"Because AR and CG are both associated with positive downward relative gradients of $Z_H$, it means that on average AR is more efficient to increase the particle size (albeit decreasing its density) than CG."

**Anonymous Referee #2**

Identification of snowfall microphysical processes using radar observations is a challenging

topic, because of the similarity of the radar signatures of some of the processes. Changes of radar variables as a function of altitude carry additional information. The authors are proposing a new method that can be used to advance our interpretation of radar observations of ice microphysical processes. The manuscript is structured well and clear. I have listed a few comments that I would like the authors to address before the paper can be accepted.

We gratefully thank Reviewer #2 for his/her review our paper. We answer his/her comments herebelow:

General comments:
1) I really like how the authors used the continuity equation to derive conditions which could be used to diagnose if vertical gradients of radar variables are driven by microphysics. Because this part is rather new, for the radar community, it would be good to explain in a few sentences what these conditions actually mean. For example, the condition 1 indicates that the reflectivity field is horizontally homogeneous.

We agree that this part should be better introduced. Following the reviewer's comment, we have added:
For Condition 1: We have added an interpretation "i.e. the velocity field is sufficiently homogeneous horizontally and the horizontal advection of $Z_H$ horizontal inhomogeneities is negligible" as well as an interpretation of this inequality in a simple (but common) case: " In stratiform snowfall conditions, the horizontal wind speed (resp. the relative fall speed velocity) is frequently 'smoother' than the radar reflectivity in the horizontal (resp. vertical) direction. In such situations, Eq. 7 approximates to:

$$\frac{\overline{U}}{L_{x,Z_H}} \ll \frac{\overline{W}}{L_{z,Z_H}}$$

And condition 1 this reduces to ensuring that the horizontal advection time scale of the reflectivity is much smaller than the vertical one."
For Condition 2: "In other words, the vertical changes of the relative vertical velocity are small compared to the changes in reflectivity".
Condition 3: This condition indicates that the system is quasi-stationary as mentioned in the text.

2) Related to the continuity equation, have you checked if the conditions 1,2, and 3 are the same if you are using reflectivity and differential reflectivity. I wonder if the underlying sensitivity of these radar variables is different.
ZDR is an extensive variable that obeys a similar continuity equation as ZH. Therefore, the comparison of the different terms in the equation and the conditions are similar. The sensitivity of the radar to both variables may indeed play a role and can be responsible for some of the differences between the values reported in Table 1 for the characteristic length scales of $Z_{DR}$ and $Z_H$ signal. That is the reason why we computed the lengths for both values (except horizontally as the model cannot provide $L_{x,ZDR}$).
We added the following sentences to make this point clearer in the manuscript lines 96-98:
"A similar continuity equation can be developed for other extensive radar variables, such as $Z_{DR}$, and therefore similar developments hold, i.e. similar conditions on the characteristic scales of variation can be derived"
"The method being based on the vertical gradients of $Z_H$ and $Z_{DR}$, it can be reliably applied to snowfall events that respect the three conditions (Eq. 7, Eq. 9 and Eq. 11) for both $Z_H$ and $Z_{DR}$."

3) - Case studies. I miss a figure showing Z and Zdr HTI figures for the case studies. I

wonder if they can be included, as a supplementary material for example.
Such figures have been included in the supplement (Figs. S3 and S4).

4) - Sublimation. This is probably the most challenging class for my understanding and
interpretation. In Fig. 5 and 6 you show the PIVS results. The SUB class is rather
prevalent and sometimes its appearance is puzzling.
For example, in Fig. 5 starting from the cloud top, PIVS shows CR then SUB then AR then SUB. I
cannot figure out how the transition from CR to SUB and then to AR could happen. It would be
very helpful to see radio sounding measurements and more discussion on what SUB actually means
in this case.

This layer of SUB is indeed surprising. However, we can notice a sub-saturated layer with respect to
ice in the radiosoundings in the new Figs. 3 and 4. Such subsaturated layer is likely responsible for
this SUB patterns during EV1. We think the addition of the radiosoundings for both events in the
manuscript will help the interpretation of the SUB.
We have modified the corresponding sentence in the text:
"Albeit surprising, the thin upper layer of SUB visible around 3000 m between 28 December at 14:00
UTC and 29 December at 09:00 UTC concurs with a sub-saturated layer with respect to ice visible in
the radiosoundings (see Fig. 3i). Occurrence of sublimation within this layer is thus very likely."

More generally, sublimation is a process that is directly dependent on the RHi field. This
thermodynamic variable is known to be heterogeneous and to vary rapidly in both time and space.
This likely explains the fragmentation of SUB in PIVS output.

Specific comments:

5) Lines 193-194: "as particles become more spherical (less oblate)" this is an incomplete
statement, also changes in particle density would affect Zdr.

We thank the reviewer for this correction. We modified the manuscript Line 204-207 with the
following:
"Aggregation and riming (hereafter AR) correspond to an increase in reflectivity due to an increase in
particle size and/or density with decreasing altitude ($\partial_z Z_H < 0$) and a decrease with decreasing
height in $Z_{DR}$ ($\partial_z Z_{DR} > 0$). This decrease in $Z_{DR}$ is due to the decrease in particles density (larger
snowflakes being less dense) and to the more chaotic orientation of the snowflakes associated with
their increase in size and Reynolds number (Li et al., 2018; Ryzhkov and Zernic, 2019). "

6) Lines 198-200: "Crystal growth by vapor deposition (hereafter CG) corresponds to
an increase in particle size with decreasing height (@z ZH < 0) and an increase in
oblateness with decreasing height (@z ZDR < 0) as particles generally grow along their
longest dimension (Schneebeli et al., 2013; Andric¸et al., 2013; Grazioli et al., 2015)" In some cases,
ice crystal growth would also change density. For example, growing
dendrites will become more oblate, but at the same time the density would decrease.
These two processes affect dual -polarization radar signatures in opposite direction. I
wonder if the statement that ZDR will increase as particle grow by vapour deposition is
valid generally? Could you clarify this point

Indeed, dendritic growth will affect ZDR in two opposite directions: (i) the increase in oblateness will
act to increase ZDR, while the decrease in "effective" density will tend to decrease ZDR (through a
decrease in the dielectric constant). However, since ZDR is the reflectivity-weighted measure of

particle shape (Kumjian 2013), the effect of increase in oblateness (which is weighted by $D^6$ in $Z_{DR}$) will dominate over the decrease in density. Hence we argue that yes this statement is valid generally. The rate of increase of ZDR (i.e. the second derivative) depends on the habit though and thus on temperature and humidity conditions. Indeed, for the same minor-to-major axis ratio, higher-density habits, such as plates will lead to faster increase of $Z_{DR}$ compared to lower-density habits such as dendrites or needles (Hogan et al. 2002, Kumjian 2013).

We precise this in the manuscript with "Crystal growth by vapor deposition (hereafter CG) corresponds to an increase in particle size with decreasing height ($\partial_z Z_H < 0$) and an increase in oblateness with decreasing height ($\partial_z Z_{DR} < 0$) as particles generally grow along their longest dimension (Schneebeli et al., 2013; Andric et al., 2013; Grazioli et al., 2015). The decrease in density during CG acts the opposite way as the increase in oblateness on $Z_{DR}$ (though the dielectric constant). However, since in $Z_{DR}$ the oblateness is weighted in $D^6$, the contribution from the increase in oblateness generally dominates over the decrease in density, and overall contributes to an increase in $Z_{DR}$ during CG. "

Hogan, R. J., Field, P., Illingworth, A., Cotton, R., and Choularton, T.: Properties of embedded convection in warm-frontal mixed-phase cloud from aircraft and polarimetric radar, Quarterly Journal of the Royal Meteorological Society: A journal of the atmospheric sciences, applied meteorology and physical oceanography, 128, 451–476, 2002

Kumijan, M. R.: Principles and Applications of Dual-Polarization Weather Radar. Part II: Warm-and Cold-Season Applications., Journal620of Operational Meteorology, 1, 2013